# Endoplasmic reticulum–plasma membrane contact gradients direct cell migration

Bo Gong[1,2 ✉], Jake D. Johnston[3,4], Alexander Thiemicke[1,2], Alex de Marco[4,5] & Tobias Meyer[1,2 ✉]

Directed cell migration is driven by the front–back polarization of intracellular signalling[1–3]. Receptor tyrosine kinases and other inputs activate local signals that trigger membrane protrusions at the front[2,4–6]. Equally important is a long-range inhibitory mechanism that suppresses signalling at the back to prevent the formation of multiple fronts[7–9]. However, the identity of this mechanism is unknown. Here we report that endoplasmic reticulum–plasma membrane (ER–PM) contact sites are polarized in single and collectively migrating cells. The increased density of these ER–PM contacts at the back provides the ER-resident PTP1B phosphatase more access to PM substrates, which confines receptor signalling to the front and directs cell migration. Polarization of the ER–PM contacts is due to microtubule-regulated polarization of the ER, with more RTN4-rich curved ER at the front and more CLIMP63-rich flattened ER at the back. The resulting ER curvature gradient leads to small and unstable ER–PM contacts only at the front. These contacts flow backwards and grow to large and stable contacts at the back to form the front–back ER–PM contact gradient. Together, our study suggests that the structural polarity mediated by ER–PM contact gradients polarizes cell signalling, directs cell migration and prolongs cell migration.

Directed cell migration is vital for processes such as organ formation, tissue repair, pathogen defence and cancer metastasis[1,2]. In response to environmental cues, single cells and collectively migrating cells polarize their signalling and actin machineries from the front to the back to establish front–back cell polarity[1,3,10]. Genetic studies have revealed severe migration defects when receptor tyrosine kinase (RTK) signalling is lost[4,11,12]. RTKs locally increase phosphatidylinositol (3,4,5)-trisphosphate lipids and activate CDC42 and Rac small GTPases at the front of migrating cells to polymerize actin, protrude membranes outwards and direct cell migration[3,5,10,13].

However, how cells generate and maintain their front–back polarity was a mystery. Theoretical and experimental work predicted the existence of a long-range inhibitory mechanism, although its precise identity was elusive[6,7,14–17]. Our study was motivated by the idea that cell migration is intricately linked to the structural polarization of the cell cortex. Indeed, our previous work revealed a polarized actin cortex network, with more membrane proximal actin filaments supporting the rear PM, which is potentially associated with membrane tension to prevent the formation of multiple fronts in migrating cells[8,9]. However, phosphatidylinositol (3,4,5)-trisphosphate and small GTPase signals and actin dynamics can change locally within seconds[18], which indicated that a longer-lasting structural polarity mechanism is needed to maintain stable polarization of the cell cortex.

## Persistent cell polarization during migration

We investigated the polarization mechanism of cells using human retinal pigment epithelial cells (RPE-1 cell line) migrating on micropatterned linear stripes of fibronectin in 96-well plates (Extended Data Fig. 1a). In the presence of uniformly distributed serum or EGF stimuli, cells rapidly migrated by randomly selecting one of the two possible directions on the fibronectin stripes (Extended Data Fig. 1b,c). Cell migration was followed over time using automated multisite confocal fluorescence microscopy by imaging the expressed PM marker CAAX–mTurquoise.

To understand how such cells polarize, we measured front–back cell polarity by fixing migrating cells and staining for phosphotyrosine residues using a pan-phosphotyrosine antibody (pan-pTyr)[19]. Despite the uniform stimulus, there was a gradient of intracellular pan-pTyr signals: high at the front and low at the back (Extended Data Fig. 1d–f). Furthermore, addition of the phosphatidylinositol 3-OH (PI3K) inhibitor LY294002 to cells migrating on stripes rapidly inhibited PI3K signalling (Extended Data Fig. 1g) and stopped cell migration[20] (Extended Data Fig. 1h). After washing out LY294002 20 min later, most cells continued migrating in the same direction rather than choosing a random direction (Extended Data Fig. 1h,i). These two observations motivated us to search for a lasting polarized structure that could persistently polarize receptor signalling.

## Polarized ER–PM contacts

Electron microscopy studies[21–23] have identified sites where the PM is attached to the ER, with a gap distance of 10–25 nm. These ER–PM contact sites can be stably maintained by extended synaptotagmin (E-Syt) proteins, and individual double-membrane contact areas have variable diameters of about 200 nm (refs. 24,25). We speculated that

[1]Department of Cell and Developmental Biology, Weill Cornell Medicine, New York, NY, USA. [2]Department of Biochemistry, Weill Cornell Medicine, New York, NY, USA. [3]Department of Physiology and Cellular Biophysics, Columbia University, New York, NY, USA. [4]Simons Electron Microscopy Center, New York Structural Biology Center, New York, NY, USA. [5]Department of Biochemistry and Molecular Biophysics, Columbia University, New York, NY, USA. ✉e-mail: bog4001@med.cornell.edu; tom4003@med.cornell.edu

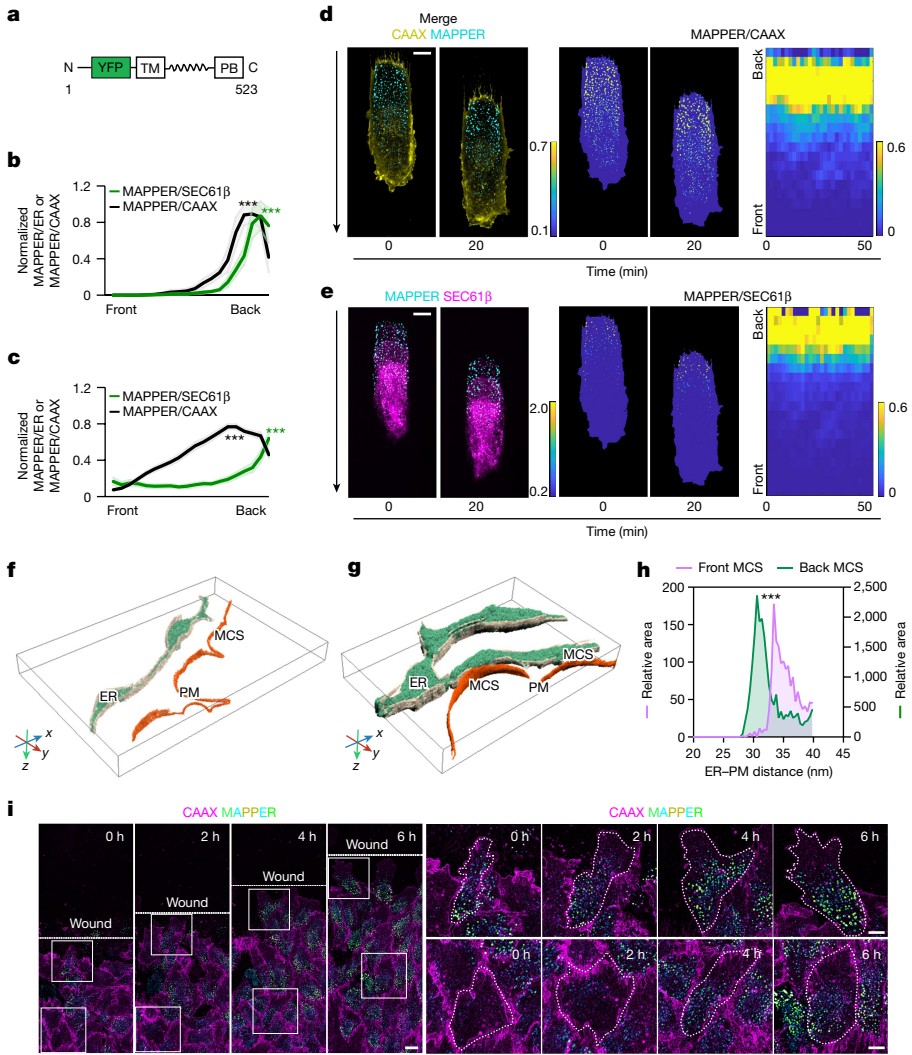

**Fig. 1 | Migrating cells have stable back-to-front gradients of the ER–PM contact density. a**, Schematic of the ER–PM contact reporter MAPPER, which is exclusively localized to the ER by its transmembrane (TM) segment and reversibly interacts with the PM through its polybasic domain (PB). **b**, Gradients of MAPPER ratios from single cells in **d** and **e** against CAAX (marker of PM) and SEC61β (marker of ER) at different times and averaged (bold). Mean ± s.d. The 20% front and back signals were compared. $P = 7.1899 \times 10^{-59}$ (black), $P = 4.2353 \times 10^{-40}$ (green). **c**, Group average MAPPER to CAAX ratio and the MAPPER to SEC61β ratio of 24 cells. Mean ± s.e.m. $P = 2.1291 \times 10^{-21}$ (black), $P = 8.6632 \times 10^{-6}$ (green). **d,e**, Left, pairs of images captured 20 min apart of representative RPE-1 cells stably expressing MAPPER–mVenus (cyan) along with mTurquoise–CAAX (yellow) (**d**) and iRFP–SEC61β (magenta) (**e**). Middle, MAPPER to CAAX ratio and MAPPER to SEC61β ratio colour images. Right, kymographs of

the average back-to-front intensity ratio over time (time interval of 2 min). Scale bar, 10 μm. **f,g**, Cryo-ET segmentation and 3D rendering of ER and PM from the front (**f**) and back (**g**) of a migrating cell. MCS indicates the site where the distance between the ER and PM is within 40 nm. **h**, Surface-area-weighted histogram of the ER–PM contact area of the front versus back ER. All ER within a 40 nm gap from the PM was included. $n = 7$ front and $n = 7$ back tomograms used for quantification. Two-tailed, paired $t$-test, $P = 1.9461 \times 10^{-5}$. **i**, Time-course of collectively migrating RPE-1 cells stably expressing mTurquoise–CAAX and MAPPER–mVenus in the wound closure assay. Right panels show magnified representative leader cells (top row) and follower cells (bottom row) respectively, from the boxed regions on the left to highlight the distribution of ER–PM contact gradients over time. The PM is marked by a white dashed boundary. Scale bar, left, 20 μm; right, 10 μm. Unpaired two-tailed $t$-test in **b,c**.

such stable ER–PM contacts may contribute to the persistence of cell polarization and polarized receptor signalling.

In mammalian cells, the cortical ER covers about 0.25–8% of the PM, forming hundreds of punctate ER–PM contacts[23,26]. We used the ER–PM contacts marker MAPPER[26] to monitor their distribution and dynamics in living cells. This reporter consists of a transmembrane domain that is anchored by an amino-terminal signal peptide in the ER, a linker sequence and a carboxy-terminal polybasic motif that reversibly binds to negatively charged lipids in the PM if it is within less than 25 nm from the ER (Fig. 1a and Extended Data Fig. 1j).

To determine whether ER–PM contact sites might be polarized in migrating cells, we used this MAPPER reporter along with markers for the ER and PM in the same cell (SEC61β–iRFP and CAAX–mTurquoise,

respectively). The density of the overall ER–PM contact area had a polarized distribution, with low ER–PM contact area density at the front and high density at the back (ER–PM contact gradient) (Fig. 1b–e and Supplementary Video 1). The same polarized distribution was observed when we referenced the ER–PM contact gradient relative to the PM or ER marker (Fig. 1b,c). Serial image analysis using a kymograph representation showed that the ER–PM contact gradient is stable in migrating cells over time (Fig. 1d,e).

To directly track the distribution of endogenous ER–PM contact sites, we conducted a total internal reflection fluorescence image analysis of endogenous E-Syt1 staining. The results confirmed a back-to-front ER–PM contact gradient at the adhesion surface (Extended Data Fig. 2a,b). We further confirmed the presence of ER–PM contact gradients by

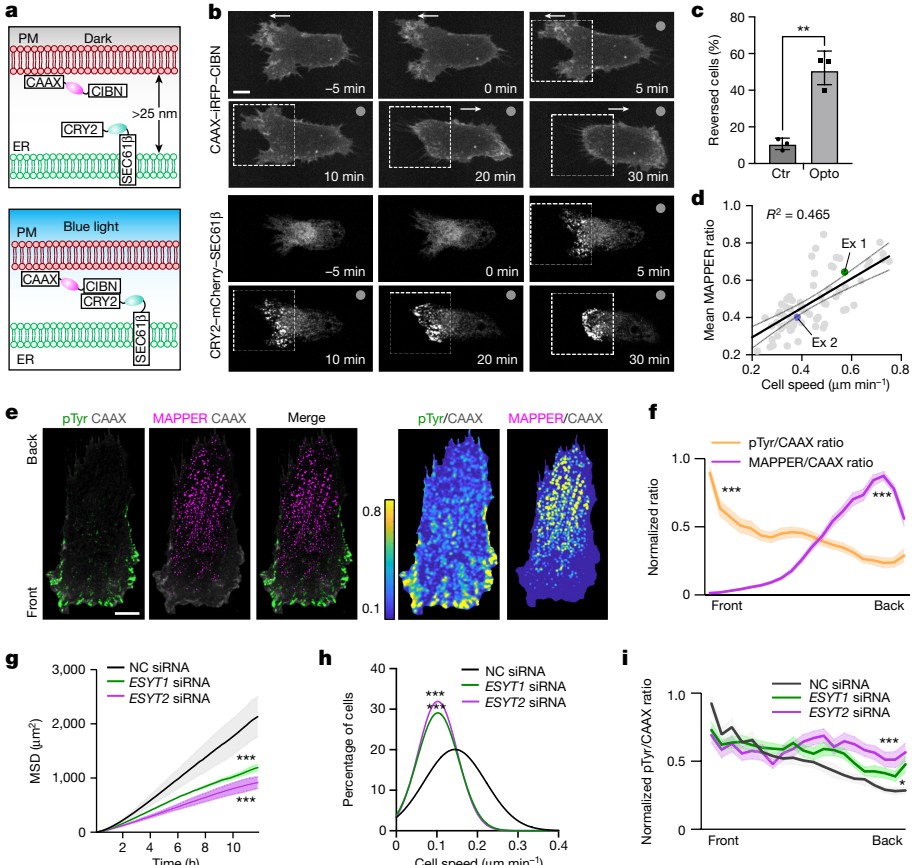

**Fig. 2 | MAPPER gradients direct cell migration by restricting pTyr signals to the front. a**, Schematic of the optogenetics tool. CRY2–mCherry–SEC61β and CAAX–iRFP–CIBN were stably expressed in RPE-1 cells to induce ER–PM contacts by local illumination. **b**, Cell movement before and during illumination (repeated every 2 s). White arrows and boxes mark the direction of movement and illuminated region, respectively. **c**, Percentage of illuminated cells that reversed the direction of migration. $n = 29$ cells in the control group (Ctr; stably expressing CAAX–iRFP and SEC61β–eGFP) and $n = 39$ cells in the optogenetic (Opto) group. $P = 0.002$, data from 3 experiments. **d**, Scatter plot of mean MAPPER to CAAX gradient steepness and mean cell speed for individual cells with fitted regression line ± 95% confidence interval ($n = 69$ cells from 3 experiments). **e**, pan-Tyr staining of migrating RPE-1 cells. Right, pTyr to CAAX ratio and MAPPER to CAAX ratio images. **f**, Mean MAPPER to CAAX gradient compared with pTyr to CAAX gradient. $n = 24$ cells. The 20% front and back signals were compared. $P = 4.3356 \times 10^{-39}$ (purple), $P = 5.955 \times 10^{-11}$ (orange). **g**, MSD analysis of RPE-1 cells transfected with control (NC), *ESYT1* or *ESYT2* siRNAs after 20 ng μl$^{-1}$ EGF stimulation. $n = 2,739$ (control), 3,658 (*ESYT1*) and 4,082 (*ESYT2*) cells. Control: $P = 2.3357 \times 10^{-43}$ (vs *ESYT1* siRNA), $P = 2.8549 \times 10^{-79}$ (vs *ESYT2* siRNA). **h**, Frequency distribution of mean cell speed from **g**. $n = 2,333$ (control), 2,291 (*ESYT1*) and 3,041 (*ESYT2*) cells. Control: $P = 7.8469 \times 10^{-97}$ (vs *ESYT1* siRNA), $P = 2.5747 \times 10^{-157}$ (vs *ESYT2* siRNA). **i**, pTyr to CAAX ratio gradient in migrating RPE-1 cells transfected with control, *ESYT1* or *ESYT2* siRNAs. $n = 10$ (control), 21 (*ESYT1*) and 14 (*ESYT2*) cells. Control: $P = 0.0488$ (vs *ESYT1* siRNA), $P = 6.1594 \times 10^{-4}$ (vs *ESYT2* siRNA). Unpaired two-tailed *t*-test (**c**,**f**) or one-way analysis of variance (ANOVA) and Scheffe's/Dunn post hoc comparison (**g**,**h**,**i** (Dunn)). Mean ± s.d. (**c**,**g**) or mean ± s.e.m. (**f**,**i**). Scale bar, 10 μm (**b**,**e**).

comparing the ER–PM contact area of the front with the back in cryo-genic electron tomography (cryo-ET) images. The analysis revealed about 12 times more ER proximal to the PM (<40 nm gap distance) in the back compared with the front ER (Fig. 1f–h).

Notably, there was also a corresponding size gradient, with large ER–PM contacts reaching area sizes of over 0.4 μm$^2$ at the back, which was in contrast to small front contacts (Extended Data Fig. 2c–e and Supplementary Video 1). To monitor ER–PM contact sizes below the resolution limit, we calculated a predicted ER–PM contact (MAPPER) mass as the product of the intensity and measured puncta area, which gradually increased from the front towards the back (Extended Data Fig. 2f–h).

The same ER–PM contact gradient was also observed in migrating fibroblast cells (BJ-5ta cell line) and in endothelial cells (HUVEC cell line) (Extended Data Fig. 2i–l). Furthermore, during collective cell migration in a wound healing assay, ER–PM contact gradients were observed in both leader and follower RPE-1 cells (Fig. 1i, Extended Data Fig. 2m,n and Supplementary Video 2). ER–PM contact gradient were also observed in single cells migrating in two dimensions (Extended Data Fig. 2o) and in a 3D migration assay, in which RPE-1 cells were migrating in an

amoeboid-like fashion (Extended Data Fig. 2p and Supplementary Video 3). We conclude that migrating cells have small ER–PM contacts at the front and large ER–PM contacts at the back, with much lower overall contact area density at the front.

## Role of ER–PM contact gradients

We next determined whether the ER–PM contact gradient regulates cell migration. We took an optogenetic approach by expressing ER-localized CRY2–SEC61β–mCherry and PM-localized CAAX–CIBN–iRFP. This system can be used to locally increase endogenous ER–PM contacts (Fig. 2a). Before local light activation, which dimerizes or oligomer-izes the ER and PM localized constructs, CRY2–SEC61β–mCherry was evenly distributed at the ER as expected (from −10 min to 0 min) and the cell was moving towards the left (Fig. 2b and Supplementary Video 4). After blue light activation in the front region, the number of CRY2–SEC61β–mCherry-positive puncta gradually increased, which indicated that ER–PM contacts were growing. Notably, along with the newly forming contacts, the membrane protrusion in the initial front gradually retracted while small new protrusions started to appear in

the original back. This process ultimately resulted in a reversal of cell migration, with the cell moving towards the right (Fig. 2b and Supplementary Video 4). Statistical analysis of this turning behaviour is shown in Fig. 2c. Thus, ER–PM contact gradients can control the direction of cell migration.

Quantitative analysis showed that the steepness of the ER–PM contact gradients differed among RPE-1 cells and was positively correlated with the speed of cell migration ($R^2 = 0.465$, $n = 69$ cells from 3 independent experiments; Fig. 2d and Extended Data Fig. 3a). Furthermore, a direct comparison of the gradients in pan-pTyr signals (Fig. 2e, green) and ER–PM contact densities (Fig. 2e, magenta) revealed an inverse correlation. This result supports the hypothesis that the steepness of ER–PM contact gradients regulates the steepness of polarized signalling, which in turn controls the speed of migration (Fig. 2e,f).

When we transfected small interfering RNAs (siRNAs) against the two E-Syt proteins expressed in RPE-1 cells (E-Syt1 (encoded by *ESYT1*) and E-Syt2 (encoded by *ESYT2*)), their protein levels were significantly reduced together with a reduced number and density of ER–PM contacts (Extended Data Fig. 3b–e). This result confirmed that E-Syt proteins have a role in ER–PM contact formation[24,25]. Consistent with the requirement of ER–PM contacts for cell migration, knockdown of *ESYT1* or *ESYT2* reduced the EGF-mediated migration speed. This was measured in a 2D migration assay by quantifying the travel distance and the mean square displacement (MSD) (Fig. 2g,h and Extended Data Fig. 3f,g). The same inhibitory effect on cell migration was also seen in serum-stimulated cell migration (Extended Data Fig. 3h,i). Notably, knocking down *ESYT1* and *ESYT2* reduced the front-to-back pan-pTyr signal gradient, which was manifested as a significant pan-pTyr signal increase at the back (Fig. 2i and Extended Data Fig. 3j). Thus, it seems that ER–PM contact gradients regulate the direction and speed of migration by suppressing pTyr signalling at the back.

## Polarized PTP1B activity

The ER-resident tyrosine phosphatase PTP1B can signal from the ER to the PM by directly interacting with substrates at the PM[27,28]. We considered whether the polarization of ER–PM contacts is making the PM-localized substrates of PTP1B at the back of cells more accessible to the ER-localized PTP1B phosphatase, which could then suppress RTK signalling at the back. Such spatially controlled phosphatase activity towards the back could explain why the signal of pTyr residues is low at the back and high at the front of migrating cells.

To test whether PTP1B has a role in polarizing pTyr distribution, we applied two selective PTP1B allosteric inhibitors: CAS765317-72-4 and MSI-1436. CAS765317-72-4 prevents closure of the WPD loop of PTP1B[29], whereas MSI-1436 targets the disordered C terminus of PTP1B and is currently in clinical trials for the treatment of obesity-related diseases[30,31]. Adding either of the PTP1B inhibitors led to even distributions of the pan-pTyr signal (Fig. 3a,b,d). Control experiments showed that treatment of cells for 25 min with the inhibitors did not significantly change the ER–PM contact signal gradient (Fig. 3c). In a second strategy to reduce PTP1B activity, we knocked down *PTP1B* using siRNAs (Extended Data Fig. 4a). This approach abolished the pTyr gradient (Extended Data Fig. 4b–d), a result consistent with the PTP1B inhibitor data. When we overexpressed PTP1B–mCherry and monitored its localization against the ER marker, it showed an even distribution in migrating cells (Extended Data Fig. 4e,f), which argued against a potential gradient in PTP1B itself along the ER. Application of the PTP1B inhibitors also inhibited cell migration, as measured either by MSD or speed analysis in 2D after EGF stimulation (Fig. 3e,f). These results suggest that PTP1B activity is polarizing RTK signalling to facilitate cell migration.

Previous work has suggested that ER-localized PTP1B can dephosphorylate one of its substrates, EGFR, at the PM[27,28]. We used a bimolecular fluorescence complementation (BiFC) assay to determine whether the interaction between PTP1B and EGFR is polarized in cells. The YC fragment (residues 155–238) of YFP (eYFP) was conjugated to the N terminus of the PTP1B substrate-trapping mutant form PTP1B(D181A)[32], and the YN fragment (residues 1–154) of eYFP was conjugated to the C terminal of EGFR (Fig. 3g). As expected, YC–PTP1B(D181A) was localized to the ER and EGFR–YN was localized to the PM (Extended Data Fig. 4g,h). As shown by the polarized and punctate eYFP signal, YC–PTP1B(D181A) and EGFR–YN specifically interacted at the back of migrating cells (Fig. 3h). Moreover, the complemented local eYFP signal colocalized with the ER–PM contact reporter (Fig. 3h and Extended Data Fig. 4i). Taken together, these results support a model whereby the ER-resident PTP1B interacts with PM-localized substrates at ER–PM contacts and therefore suppresses RTK signalling selectively at the back as there is a much higher density of ER–PM contacts at the back.

## Retrograde ER–PM contact flow

Like cortical actin[33,34], individual ER–PM contacts in migrating cells were mostly stationary relative to the extracellular matrix (Fig. 4a and Supplementary Video 5), which is contrasted by the net forward movement of the cell itself (Supplementary Video 5). Thus, from the perspective of migrating cells, actin filaments and ER–PM contacts both undergo retrograde flow (Fig. 4b).

We used two different scenarios to test how ER–PM contact gradients dynamically change when cells undergo depolarization and repolarization. We first focused on cells that temporally stop in the middle of stripes and then resume their movement by first applying and then washing out LY294002 (a PI3K inhibitor) 20 min later (as in Extended Data Fig. 1h). Even though the cells almost immediately stopped, the ER–PM contact gradient was gradually reduced but not eliminated by the temporary addition of the PI3K inhibitor (Extended Data Fig. 5a,b, the maximal front-to-back ratio was around 0.8). After the inhibitor was washed out, both the ER–PM contact gradient steepness and migration speed recovered, with cells resuming migration in the same direction (Extended Data Fig. 5a,b). Thus, the observed persistence in migration direction after cells temporally stop could be explained by the persistence of the orientation of the ER–PM contact gradient.

Second, we analysed cells that reversed their migration direction after reaching the end of stripes (Fig. 4c and Supplementary Video 6). Kymograph and time-course analysis showed a marked reorganization of MAPPER gradients during the paused phase when cells reached the end before the turn (Fig. 4c–f, from $t = 20$ min to $t = 60$ min). During this pause, the ER–PM contacts stayed in place as the density gradually decreased at the original back owing to cell contraction (Fig. 4c and Extended Data Fig. 5c,d,e, indicated by asterisks). Along with the reversal and initiation of new protrusions at the former back (Fig. 4c, protrusion happened at $t = 60$ min), the ER–PM contact density kept increasing at the former front (Fig. 4c, indicated by arrows). Meanwhile, remaining ER–PM contacts at the former back flowed retrograde towards the new back (Fig. 4c, indicated by asterisks). Thus, the slow dynamics of local ER–PM contact formation and flow can explain the slow reversal of the ER–PM contact gradient and persistence of directed cell migration.

To directly compare the growth rate of ER–PM contacts at the front with the back of migrating cells, we exploited the fact that ER–PM contacts are stationary relative to the extracellular matrix (Fig. 4a,g and Supplementary Video 5). A plot of the MAPPER to CAAX ratio as a function of the relative ER–PM contact position (Fig. 4h (0 indicates the cell front, whereas 1 indicates the back) and Extended Data Fig. 5f) showed that the ER–PM contact density increased slowly in the front half of the cell but had a threefold higher growth rate in the back half (Fig. 4h,i and Extended Data Fig. 5g). This result implied that the growth of ER–PM contacts is suppressed at the front (Fig. 1b–e and Extended Data Fig. 2c–h).

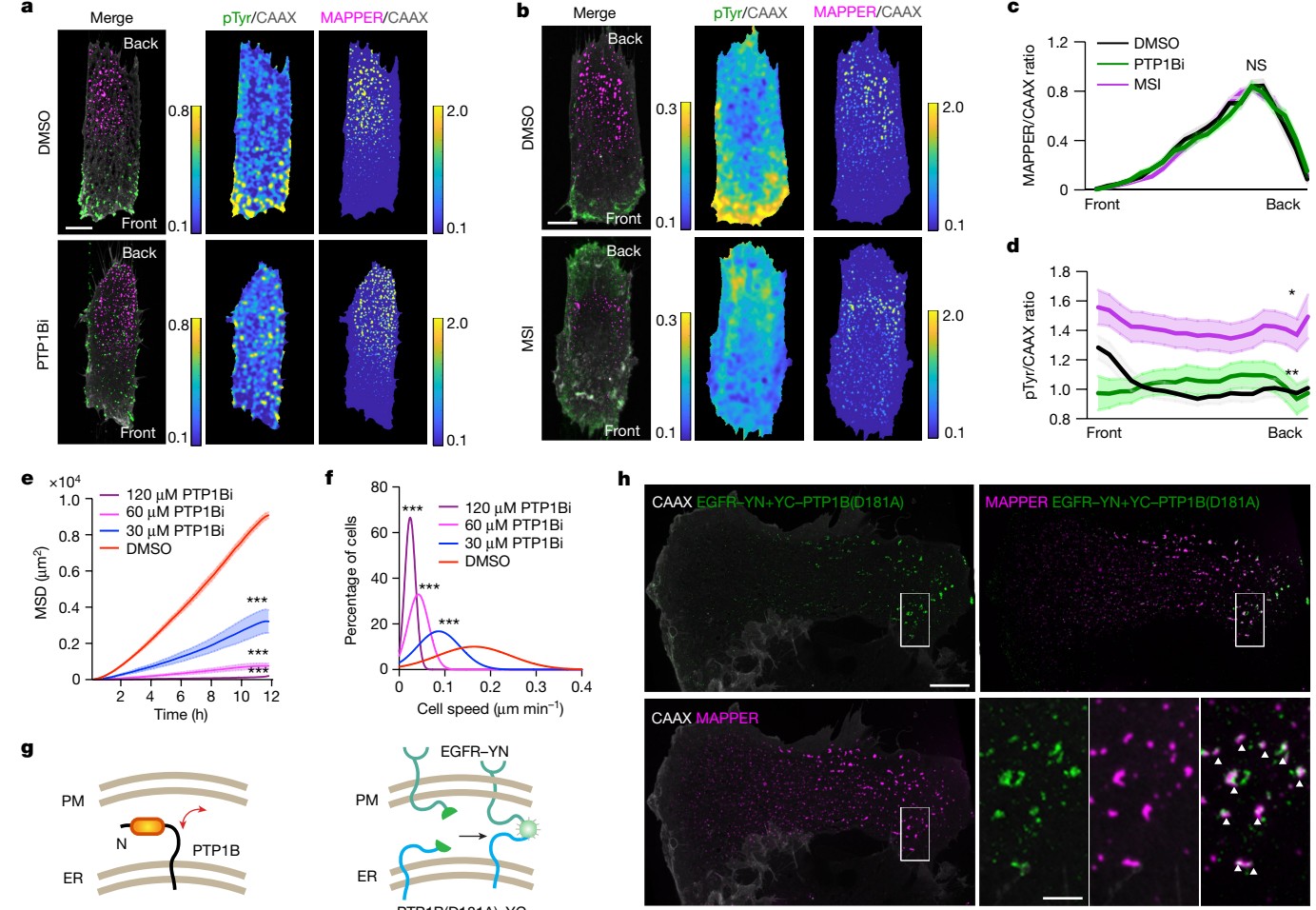

**Fig. 3 | PTP1B-mediated phosphatase activity is required for asymmetric pTyr signalling. a,b**, Representative MAPPER to CAAX and pTyr to CAAX ratio images of RPE-1 cells treated with 10 μM DMSO or the PTP1B inhibitor CAS765317-72-4 (PTP1Bi) (**a**) or the alternative PTP1B inhibitor MSI-1436 (MSI) (**b**) (treated for 25 min before fixation). Normalized scale. **c,d**, Comparison of MAPPER to CAAX (**c**) and pTyr to CAAX ratio (**d**) gradient in RPE-1 cells treated with DMSO, PTP1Bi or MSI. Mean ± s.e.m. $n$ = 16 (DMSO), 29 (PTP1Bi) and 34 (MSI) cells. In **c**, DMSO: $P$ = 0.8898 (vs PTP1Bi), $P$ = 0.4472 (vs MSI). NS, not significant. In **d**, DMSO: $P$ = 0.0094 (vs PTP1Bi), $P$ = 0.0353 (vs MSI). **e,f**, MSD (**e**) and averaged speed distribution (**f**) of cells stimulated with 20 ng μl⁻¹ EGF along with DMSO or PTP1Bi at indicated concentrations. Mean ± s.d. In **e**, $n$ = 3,158 (DMSO), 2,096 (30 μM PTP1Bi), 2,211 (60 μM PTP1Bi) and 2,237 (120 μM PTP1Bi)

cells. DMSO: $P$ = 2.9294 × 10⁻³⁰⁷ (vs 30 μM PTP1Bi), $P$ = 2.0482 × 10⁻³¹¹ (vs 60 μM PTP1Bi), $P$ = 0 (vs 120 μM PTP1Bi). In **f**, $n$ = 3,458 (DMSO), 2,875 (30 μM PTP1Bi), 2,589 (60 μM PTP1Bi) and 3,090 (120 μM PTP1Bi) cells. DMSO: $P$ = 0 (vs 30, 60 and 120 μM PTP1Bi). **g**, Schematic of ER-localized PTP1B interaction with PM-localized substrates. A BiFC assay was used to determine potential interactions at ER–PM contact sites between ER-localized PTP1B (D181A mutation, C-terminal YC tag) and PM-localized EGFR (with a N-terminal YN tag). **h**, Representative images of local PTP1B–EGFR complementation signals and MAPPER signals in RPE-1 cells. Boxed regions are magnified to show colocalized signals (white arrowheads). One-way ANOVA and Scheffe's/Dunnett post hoc comparison (**c,e,f,d** (Dunn)). Scale bars, 10 μm (**a,b,h**) and 2.5 μm (**h**, boxed region).

Furthermore, visual inspection of the MAPPER puncta in migrating cells suggested that ER–PM contacts at the front of migrating cells are short-lived (Supplementary Video 5). Indeed, a time-course analysis showed that ER–PM contacts stochastically appeared and disappeared on average over approximately 2 min (Extended Data Fig. 5h). A plot of the ER–PM contact-site lifetime with MAPPER mass showed a gradual increase in lifetime as the ER–PM contact mass increased. MAPPER puncta larger than about 0.2 μm² were more stable (Extended Data Fig. 5i). Thus, the smaller size and lower density of individual ER–PM contacts at the front could be caused by their instability and slower growth rate.

## Polarized ER curvature during migration

We considered that the low stability and growth rate of ER–PM contacts at the front may result from global polarization of the ER curvature. The ER has a dynamic and complex morphology, whereby tubules

with diameters of about 60 nm are more peripherally localized and flat sheet-like structures are often more perinuclear[35–37]. The reticulon member RTN4 is a ubiquitous curvature-shaping protein that is enriched in curved ER[36], whereas CLIMP63 serves as a luminal ER spacer and is enriched in flattened ER sheets[35] (Fig. 5a). We used fluorescently tagged RTN4 and CLIMP63 proteins as markers to investigate the spatial distribution of curved and flattened ER in migrating cells. We expressed low levels of doxycycline-inducible RTN4–mCherry or CLIMP63–mCherry to avoid ER morphology changes caused by overexpression. We measured the relative distribution of these proteins along with MAPPER–mVenus, the PM marker CAAX–mTurquoise and the ER membrane marker SEC61β–iRFP.

As shown in Fig. 5b, RTN4-marked curved ER tubules were significantly enriched at the front of migrating cells. Ratio images of RTN4 to CAAX and of RTN4 to SEC61β showed a clear spatial gradient from the front to the back (Fig. 5b and Extended Data Fig. 6a,c,d,g). By contrast, CLIMP63-marked flattened ER sheets were significantly enriched at the

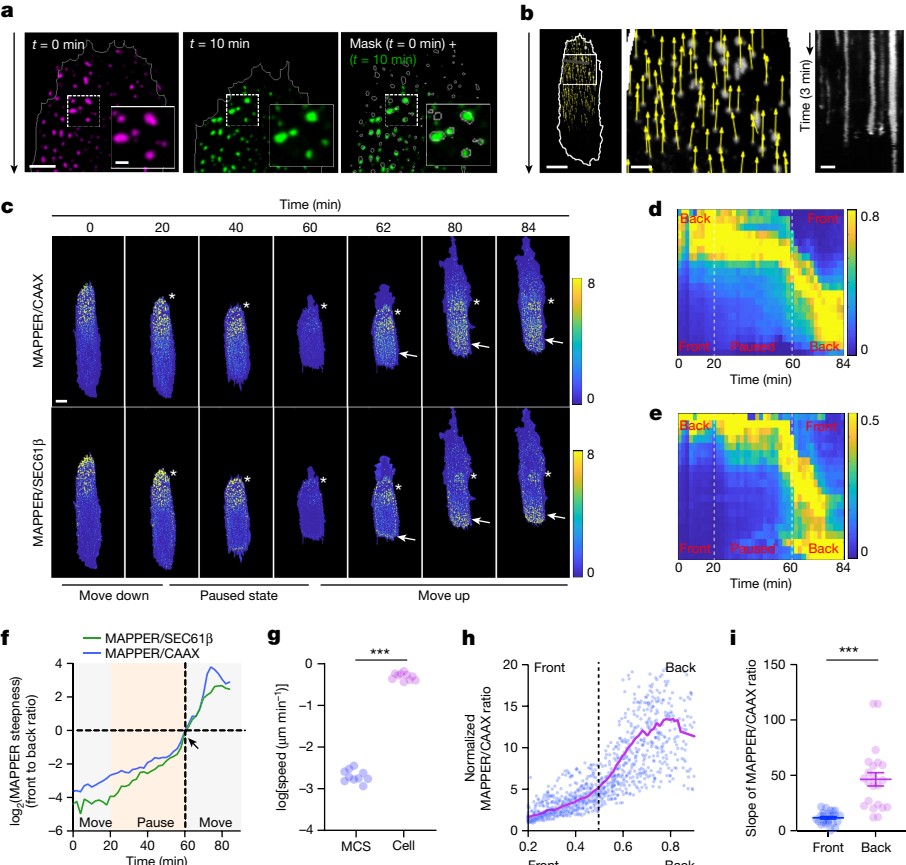

**Fig. 4 | Retrograde flow of ER–PM contacts and slow contact growth rate in front of migrating cells. a**, Representative images of MAPPER distribution (representing ER–PM contacts) stationary to the extracellular matrix at $t = 0$ min and $t = 10$ min in migrating RPE-1 cells. Grey circles, puncta masks at $t = 0$ min. **b**, Left and middle, flow field displacement vectors (yellow) of MAPPER puncta relative to the migrating cell (white, cell boundary). Length of yellow arrows denotes speed. Right, kymograph of ER–PM contact stability from the green line on the left image. **c**, Representative time-course images of the MAPPER to CAAX ratio and the MAPPER to SEC61β ratio in RPE-1 cells reversing their migration direction at the end of stripes. White asterisks, original back. Arrows, future back. $n = 53$ cells. **d,e**, Kymographs of MAPPER to CAAX (**d**) and MAPPER to SEC61β (**e**) changes of cells in **c**. $t = 20$ min to $t = 60$ min, paused window before a turn. **f**, MAPPER gradient profile change during a

turn. MAPPER gradient steepness is in $\log_2$ scale. Stalling window in yellow. Moment of reversal marked by arrow. **g**, Quantification of MAPPER puncta displacement relative to the extracellular matrix. Each dot represents the average speed of all MAPPER puncta per cell (blue) or average cell speed (pink) in a $\log_2$ scale. $P = 7.4061 \times 10^{-20}$. **h**, Growth rate of MAPPER puncta in front versus back. The MAPPER to CAAX intensity ratio of every punctum is plotted against its position in the cell ($n = 1,149$ puncta). Dashed line, middle of cell. **i**, Averaged slopes of growth rates of ER–PM contacts in front versus back of migrating cells. Fitted from bold curve in **h** by linear regression of the front and back halves. $n = 25$ cells. Mean ± s.e.m. $P = 4.9071 \times 10^{-7}$. Scale bars, 2 μm (insets in **a,b**), 4 μm (**a**) or 10 μm (**b,c**). $n = 99,710$ MAPPER puncta from 10 cells used for analysis (**a,b,g**). Unpaired two-tailed Student's $t$-test (**g,i**).

back. Quantification of the CLIMP63 to CAAX ratio and the CLIMP63 to SEC61β ratio showed a gradient in the same direction as the ER–PM contact gradient in opposition of the RTN4 gradient (Fig. 5c and Extended Data Fig. 6b,e,f,h). Control experiments showed the same direction of the back-to-front gradient using endogenous CLIMP63 immunostaining (Extended Data Fig. 6i,j).

Reconstruction and segmentation of cryo-ET images of wild-type cells confirmed that there are more ER tubules at the front and more flattened ER sheets at the back (Fig. 5e,f and Supplementary Videos 7 and 8). The surface morphometric analysis (which measures a curvature parameter) further revealed that ER at the front is about 2.23-fold more curved than at the back (Fig. 5g), a result consistent with the visually observed differences between the front and back ER. Together, these data argue that migrating cells orient the curvature of the ER network, with highly curved ER tubules enriched at the front and flattened sheet-like ER towards the back.

We next tested whether the ER membrane curvature gradient accounts for the ER–PM contact gradient. Because RTN4 and CLIMP63 are distributed in ER regions with different curvature, we used the

RTN4 to CLIMP63 ratio as a measure of the ER curvature gradient. In support of the hypothesis that ER curvature gradients are important, the shape of the front–back RTN4 to CLIMP63 ratio closely matched the inverted shape of the ER–PM contact gradient (Fig. 5d). We directly tested the role of the ER curvature gradient by knocking down *RTN4* or *CLIMP63* using siRNA. This approach enabled the manipulation of the RTN4 to CLIMP63 ratio by reducing the relative ratio and segregation of curved and flattened ER (Extended Data Fig. 7a,b). *CLIMP63* knockdown reduced the density and size of ER–PM contacts, a result consistent with the role that CLIMP63-enriched ER has in supporting the formation of ER–PM contacts (Fig. 5h and Extended Data Fig. 7c in green). By contrast, *RTN4* knockdown resulted in much larger ER–PM contacts, a result consistent with that role that tubular ER has in suppressing ER–PM contacts (Fig. 5h and Extended Data Fig. 7c in pink). In both cases, the steepness of the gradient of ER–PM contacts was reduced along with the speed of migration, which supported the hypothesis that it is not the size of the contacts but the gradient of ER–PM contacts that promotes cell migration (Fig. 5h,i and Extended Data Fig. 7d,e).

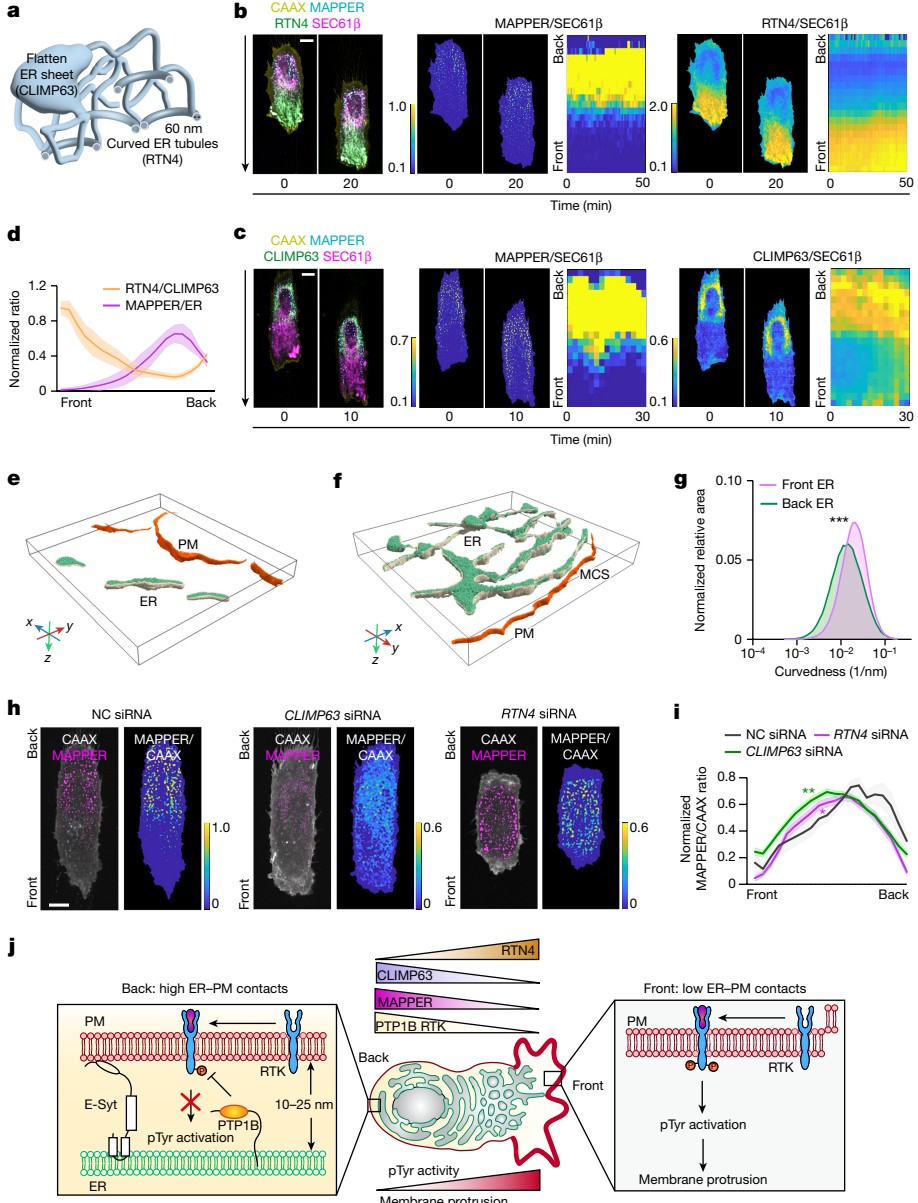

**Fig. 5 | Control of ER–PM contact gradients through a polarized organization of curved and flattened ER. a**, Schematic of curved and flattened ER morphology. Curved ER, marked by RTN4 (shown as tubules). Flattened ER, marked by CLIMP63 (shown as sheet-like). **b,c**, Distribution of doxycycline-induced RTN4–mCherry (green in **b**) or CLIMP63–mCherry (green in **c**) and SEC61β–iRFP (magenta) in migrating RPE-1 cells on stripes. Merged composite images, ratio images and kymograph analysis are shown. Time interval, 2 min. Doxycycline (1 μg ml⁻¹) was added 4 h (RTN4) and 2 h (CLIM63) before live-cell imaging. **d**, Comparison of the gradients of the mean RTN4 to CLIMP63 ratio (as a measure of the ER curvature) and the MAPPER to SEC61β ratio. The RTN4 to CLIMP63 gradient was calculated from the mean RTN4 to SEC61β gradient ($n$ = 37, 603 and 872 cells) and the mean CLIMP63 to SEC61β gradient ($n$ = 72, 541 and 349 cells). The MAPPER to SEC61β gradient was calculated from $n$ = 109, 1,144 and 1,266 cells. Mean ± s.e.m. **e,f**, Cryo-ET segmentation and 3D rendering of ER and PM from the front (**e**) and back (**f**) of cells migrating on stripes. MCS indicates the site where the distance between the ER and PM is within 40 nm. **g**, Surface morphometric analysis of front and back ER, comparing the membrane curvature. $n$ = 7 front and $n$ = 7 back tomograms, same dataset as Fig. 1f,g. Unpaired two-tailed Student's $t$-test, $P$ = 0.0006. **h**, MAPPER to CAAX ratio images of RPE-1 cells transfected with control (NC), *RTN4* or *CLIMP63* siRNAs (for 48 h). **i**, Normalized MAPPER to CAAX profiles in cells treated with control, *RTN4* or *CLIMP63* siRNAs. Mean ± s.e.m. $n$ = 12 (control), 56 (*RTN4*) and 58 (*CLIMP63*) cells. One-way ANOVA and Scheffe's post hoc comparison. Control: $P$ = 0.0025 (vs *CLIMP63* siRNA), $P$ = 0.0175 (vs *RTN4* siRNA). **j**, Model of how ER–PM contact gradients generate the observed gradient of pTyr signalling. Scale bars, 10 μm (**b,c,h**).

Microtubules (MTs), along with the motor proteins kinesin, dynein and the attachment of MT plus ends to ER proteins, position ER in the cell and help form tubular ER[38–40]. To examine which of these regulators contributes to the generation of the ER–PM contact gradient, we transfected cells with siRNAs targeting the motor protein KIF5B, the subunit p150 of the dynein motor complex (DCTN1), and EB1 to disrupt the EB1-mediated MT plus end connection to tubular ER.

Knockdown of *KIF5B* markedly reduced the number of ER–PM contact gradients (Extended Data Fig. 7f,g; the back to front ratio is around 1.6 versus 4.6 in the control siRNA group). By contrast, knockdown of *DCTN1* did not have a significant effect, whereas knockdown of *EB1* had a smaller effect on the ER–PM contact gradients (Extended Data Fig. 7f,g; the back to front ratio was 2.8 versus 4.6 in the control siRNA group). Thus, the gradient in tubular ER in these cells is probably

generated by KIF5B-mediated transport, with a smaller contribution from EB1.

In addition, CLIMP63, P180 and KTN1, three perinuclear ER-localized sheet ER proteins, can interact with different types of MTs[41], which suggests that MTs may have additional roles in stabilizing the flattened ER at the back. Indeed, *KTN1* knockdown reduced the ER–PM contact gradient, similar to *CLIMP63* knockdown (Fig. 5h,i and Extended Data Fig. 7h,i), whereas knockdown of *P180* did not result in a significant change (Extended Data Fig. 7h,i). Even though KTN1 and CLIMP63 may not directly regulate ER sheet morphology[35], these results support the idea that polarized organization of the ER requires MT binding with ER sheet proteins at the back[41].

We also determined whether ER–PM contact gradients are correlated with the orientation of the MT organizing centre, which regulates cell migration in some contexts[42]. In RPE-1 cells migrating along linear tracks, there was no clear preference of whether the MT organizing centre was at the front or the back of the nucleus, even though both orientations exhibited a similar back-to-front gradient in ER–PM contacts (Extended Data Fig. 7j–l).

Because ER organization is also closely associated with ER homeostasis, we tested whether vesicular transport from the ER or protein folding in the ER are crucial for ER–PM contact gradients. Both brefeldin A, which inhibits vesicular transport to the Golgi, and tunicamycin, which causes an accumulation of misfolded membrane proteins and ER stress, blocked cell migration and reduced the number of ER–PM contact gradients (Extended Data Fig. 8a–f). Finally, we tested for the relevance of mitochondrial ATP synthesis and confirmed that the addition of the inhibitor oligomycin compromised RPE-1 cell migration but without affecting the ER–PM contact gradient (Extended Data Fig. 8g–i).

Together, these results show that the organization and relative composition of the ER is crucial for cell migration. That is, flattened sheet-like ER favours the formation of ER–PM contacts, whereas the high membrane curvature of ER tubules in the front of migrating cells explains the small size and instability of ER–PM contacts at the front. Our results further suggest that it is the ER curvature gradient that controls the steepness of the ER–PM contact gradient, which in turn controls the signalling gradient, migration speed and direction of migration.

## Conclusions

Our study identified a gradient of the ER–PM contact density and size in migrating cells (Fig. 5j). The steepness of the ER–PM contact gradient was proportionate to the speed of migration, and loss of the ER–PM contact gradients inhibited cell migration. Unlike the signalling and actin machineries that can change rapidly and locally, ER–PM contact gradients are more stable and provide a memory of the previous migration direction by being only gradually lost after migration stops. Furthermore, stalled cells start to migrate in a new direction only after reorienting their ER–PM contact gradient towards the new back. Our study argues that there are two crucial mechanisms that ensure stable ER–PM contact gradients in migrating cells: (1) the retrograde flow of growing ER–PM contacts and (2) the growth rate differences of ER–PM contacts at the front and back conferred by an ER curvature gradient.

In addition to our finding of a PTP1B-mediated suppression of RTK signalling at ER–PM contacts in the back, ER–PM contact gradients may have additional roles in polarizing the signalling machinery and directing cell migration. For example, the higher ER–PM contact density at the back may explain the higher basal $Ca^{2+}$ level at the back of migrating immune, epithelial and other cells[19]. Long-term $Ca^{2+}$ gradients in these cells are primarily regulated by STIM–ORAI-mediated $Ca^{2+}$ influx, which occurs exclusively at ER–PM contacts[21], and we now show that these ER–PM contact sites where $Ca^{2+}$ enters the cell are enriched at the back. Higher $Ca^{2+}$ levels at the back promotes myosin activation and actin filament contraction and therefore helps cells orient the

back of cells[17,43]. ER–PM contacts are also sites of local transport of phosphatidylinositol and cholesterol lipids, and the polarization of phosphoinositides also contributes to the polarization of cells during cell migration[22,44–48]. It will be interesting to learn whether ER–PM contact gradients contribute to the polarization of phosphoinositides. Together, our data indicate that ER–PM contact gradients direct cell migration by restricting receptor signalling to the front and prevent additional fronts from forming at the back for persistent polarization.

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

## Methods

### Cell culture

hTERT RPE-1 cells (here referred to as RPE-1) (American Type Culture Collection (ATCC), CRL-4000), HUVEC/TERT2 cells (ATCC, CRL-4053), BJ-5ta cells (ATCC, CRL-4001) and 293T cells (Takara Bio, 632180) were used in this study. RPE-1 cells were maintained in DMEM/F-12, HEPES, no phenol red medium (Gibco, 11039047) supplemented with 10% FBS (Millipore Sigma, F1435). HUVEC/TERT2 cells were cultured in EBM2 basal medium (Lonza, CC-3156) supplemented with EGM2 (Lonza, CC-4176). BJ-5ta fibroblast cells were cultured in a mixture medium with 4 parts DMEM (Thermo Fisher, 11995073) and 1 part Medium 199 (Thermo Fisher, 11150059), supplemented with 10% FBS. 293T cells were maintained in DMEM (Thermo Fisher, 11995073) with 10% FBS for lentivirus preparation. RPE-1 cells were used for most experiments in this study either by generating stable cell lines or transient transfection. HUVEC/TERT2 cells and BJ-5ta cells were used for stable cell line generation for measuring ER–PM contact gradients. All stably transfected cell lines were cultured identical to the original cell line.

### Alveole/PRIMO system-based micropatterning

The Alveole/PRIMO system was used to pattern linear stripes on 96-well glass-bottomed plates (Cellvis, p96-1.5H-N) using a standard micropatterning protocol. In brief, a 96-well plate was subjected to plasma cleaning for 2 min and coated with 60 μl poly-L-Lysine (Sigma, P4707) per well for 30 min. After 3 wash steps with DI water (Invitrogen, 10977023), the 96-well plate was heated at 90 °C until dry. Meanwhile, 50 μl 100 mg ml$^{-1}$ of mPEG-SVA (Laysan Bio, MPEG-SVA-5000-5g) was freshly prepared in 0.1 M HEPES (pH 8.5) and added to each well for 1 h to make the glass non-adhesive for cells (passivation). The plate was washed 4 times using DI water and again dried to complete passivation. At this stage, the coated plated could be stored at 4 °C for several weeks. Before experiments were performed, a gel mixture (978.6 μl DI water, 20 μl PLPP gel (Alveole, B002) and 1.4 μl surfactant (Alveole, B002)) was freshly made and applied to individual wells in the 96-well plate (50 μl per well), and the 96-well plate was put on a heat plate at 90 °C. Once the gel solutions were evaporated, the 96-well plate was moved to a microscope connected to a PRIMO system for photopatterning. The HCS wizard of the Leonardo software was used to automatically control the position of each well, and a previously designed 20-μm wide linear stripe template was imported for laser patterning in each well. After photopatterning, the 96-well plate was immediately washed with 1× PBS 3 times and incubated with 70% ethanol for several minutes, followed by 4 times wash with 1× PBS. Finally, the 96-well plate was coated with 50 μl 10 μg ml$^{-1}$ fibronectin (Sigma, F1141-5mg) per well for 30 min before 1× PBS rinse. A total of 2,000 cells were seeded into each well 12 h before each experiment.

### Stable cell line generation

Stable cell lines were generated by lentivirus infection combined with cell sorting or puromycin selection. In brief, plasmids of interest in a lentiviral transfer plasmid backbone were transfected into low-passage 293T cells together with third-generation lentiviral packaging plasmids, including pMDLg/pRRE (Addgene, 12251), pRSV-rev (Addgene, 12253) and pCMV-VSV-G (Addgene, 8454), using Lipofectamine 2000 (Thermo, 11668019). The viral supernatants were collected at 48 h and 72 h after transfection and pooled together for subsequent filtration using a 0.22 μm filter (Millipore, SCGP00525) and concentrated using a 100 kDa centrifugal filter (Millipore, UFC910024). The concentrated virus was then aliquoted into several cryotubes and stored at −80 °C for future use or directly added to cells in the growth medium with polybrene (EMD Millipore, TR-1003-G). To generate RPE-1 cells with constitutive expression of mTurquoise–CAAX and MAPPER–mVenus, single cells with both fluorescence constructs after virus infection were sorted into individual wells of a 96-well plate and cultured for expansion.

After confirming that five clones showed the same observation of the back-to-front ER–PM contact gradient, one of them was chosen for most studies and used as the base cell line to generate other cell lines. We selected cells that expressed this ER–PM contact reporter at a low level to minimize its effect on cell morphology and cell polarization. pLV-iRFP–SEC61β and pLV-PTP1B–mCherry were respectively or simultaneously introduced into stable cell line with mTurquoise–CAAX and MAPPER–mVenus to construct a 3-colour stable cell line (mTurquoise–CAAX, MAPPER–mVenus and iRFP–SEC61β) or a 4-colour stable cell line, (mTurquoise–CAAX, MAPPER–mVenus, PTP1B–mCherry and iRFP–SEC61β). pLV-AKT–PH–mCherry was introduced into mTurquoise–CAAX/MAPPER–mVenus stable cell line to generate mTurquoise–CAAX/MAPPER–mVenus/AKT–PH–mCherry stable cell line. For the two doxycycline-inducible cell lines, pCW-RTN4–mCherry or pCW-CLIMP63–mCherry plasmid was introduced into the 3-colour stable cell line, mTurquoise–CAAX/MAPPER–mVenus/iRFP–SEC61β. Doxycycline (1 μg ml$^{-1}$) was added to induce RTN4 or CLIMP63 expression at the start of imaging.

### siRNA and plasmid transfection

siRNAs from Dharmacon (Supplementary Table 1) were dissolved in Ultrapure DNase/RNase free distilled water (Fisher Scientific, 10-977-023) to prepare 2 μM siRNA stock. Stock solutions were aliquoted into multiple tubes to avoid repeated thawing. For siRNA transfection experiments in 96-well plates, RPE-1 cells were seeded 16 h before transfection and transfected with siRNA using DharmaFECT 1 (Dharmacon, T-382 2001-03) according to the manufacturer's protocol. In brief, 20 nM siRNA and 0.5 μl DharmaFECT1 were diluted with Opti-MEM medium (Gibco, 31-985-070) to prepare a 10 μl volume system in separate tubes. After 5 min of incubation at room temperature, the two tubes were mixed thoroughly and gently for another 20-min incubation. Then another 80 μl Opti-MEM medium was added into the transfection mixture and transferred to each well for a 6 h transfection before the medium was replaced with the complete growth medium.

For transient transfection of DNA, Lipofectamine 2000 was used following standard protocols. In brief, 2–3 × 10$^3$ RPE-1 cells were plated per well and transfected with 0.1–0.2 μg DNA of each plasmid and 0.25 μl Lipofectamine 2000 diluted in Opti-MEM medium. The transfection mix was replaced after 4 h with complete growth medium and cells were imaged 16–24 h after transfection.

### DNA plasmids and chemicals

The following plasmids were ordered from Addgene: GFP–MAPPER (117721), iRFP–SEC61β (108125), pPTP1BD181A–mCherry (40270), pHAGE2–mCherry–RTN4a (86683), mCherry–CLIMP63 (136293), pBiFC–VN173 (22010), pBiFC–VC155 (22011), EGFR–GFP (32751) and pcDNA3.1–AKT–PH–mCherry (67301). These plasmids were used as templates to amplify required fragments, which were further assembled into the destination plasmid backbone using the Gibson assembly method (NEB, E2611L). In brief, pLV–MAPPER–mVenus, pLV–iRFP–SEC61β, pLV–PTP1B–mCherry and pLV–AKT–PH–mCherry plasmids were constructed based on the pLV–mTurquoise–CAAX backbone. After cutting this plasmid with AgeI/NotI, several fragments, including mVenus, MAPPER and FKBP for pLV–MAPPER–mVenus, iRFP–SEC61β for pLV–RFP–SEC61β, PTP1B and mCherry for pLV-PTP1B–mCherry as well as AKT–PH–mCherry for pLV–AKT–PH–mCherry, were ligated to replace the mTurquoise–CAAX insert using the recombinant cloning method. mCherry–RTN4a and mCherry–CLIMP63 were respectively amplified from their template plasmids and inserted into the pCW backbone (derived from pCW-Cas9, a gift from E. Lander and D. Sabatini, Addgene, plasmid 50661) to generate pCW–mCherry–RTN4a and pCW–mCherry–CLIMP63 plasmid. EGFR–YN plasmid was engineered from the EGFR–GFP plasmid by replacing GFP with the VN173 fragment, which was amplified from pBiFC–VN173. The PTP1B(D181A)–YC plasmid was constructed based on the pPTP1BD181A–mCherry plasmid

by replacing mCherry with the VC155 fragment, amplified from the pBiFC–VC155 plasmid.

Drugs used in the study were dissolved into DMSO (Santa Cruz, sc-358801) to prepare stock solutions, including the PI3K inhibitor LY294002 (Cayman, 70920), the PTP1B inhibitors CAS765317-72-4 (EMD Millipore, 539741-5mg) and MSI-1436 (MedChem Express, HY-12219A), and doxycycline hyclate (Sigma, D9891). All drugs were handled according to their datasheets and aliquoted to avoid repeated thawing process. The working concentrations for each drug are indicated in the corresponding experiments.

### Antibodies and Immunofluorescence
pTy (P-Tyr-1000) multiMab rabbit monoclonal antibody mix (1:500, Cell Signaling Technology, 8954), anti-ESYT1 antibody (1:200, Sigma, HPA076926) and CLIMP63 monoclonal antibody (G1/296) (1:500, Enzo Life Sciences, ENZ-ABS669-0100) were used as primary antibodies for immunostaining experiments. Secondary antibodies included goat anti-rabbit IgG(H+L) Alexa Fluor 568, Invitrogen (1:2,000, Thermo Scientific, A-11011), goat anti-rabbit IgG(H+L) Alexa Fluor 647, Invitrogen (1:2,000, Thermo Scientific, A-21245) and goat anti-rabbit IgG(H+L) Alexa Fluor 700, Invitrogen (1:2,000, Thermo Scientific, A-21038). Anti-ESYT2 antibody (1:1,000, Sigma, HPA002132), PTP1B antibody (1:1,000, BD Bioscience, 610139) and RTN4 antibody (1:1,000, Thermo Scientific, MA5-32763) were used as primary antibodies for western blotting experiments.

Cells were seeded in a 96-well glass-bottomed plate with pre-patterned linear stripes for immunostaining experiments. After siRNA transfection or inhibitor treatment, cells were fixed using 4% paraformaldehyde in PBS for 10 min at room temperature and washed with PBS. Cells were then permeabilized with 0.1% Triton-X 100 for 10 min, followed by a PBS wash and blocking buffer incubation for 1 h (10% FBS, 1% BSA, 0.1% Triton X-100 and 0.01% NaN$_3$ in PBS). Cells were then incubated with primary antibodies overnight in blocking buffer at 4 °C, followed by a PBS wash and secondary antibody incubation for 1 h at room temperature. Cells were washed with PBS again before imaging.

### Microscopy
**Automated epifluorescence microscopy.** Automated epifluorescence microscopy was used to perform live-cell time-lapse imaging to track cell migration (Figs. 2g,h, 3e,f and Extended Data Figs. 3c, f–i and 7d,e). Cells were seeded in a 96-well glass bottomed plate, coated with collagen (Advanced Biomatrix, 5005-B, 30 µg ml$^{-1}$ dilution for at least 1 h) and stained with 0.1 µg ml$^{-1}$ Hoechst 33342 (Invitrogen, H3570) in growth medium or Opti-MEM medium for 30 min at 37 °C immediately before imaging. The 96-well plates were transferred into a live-cell chamber with 37 °C, 5% CO$_2$ environment for 24 h long-term imaging by a Ti2-E inverted microscope (Nikon) equipped with a LED light source (Lumencor Spectra X) and Hamamatsu ORCA-Flash4.0 V3 sCMOS camera. The following acquisition parameters were used: interval, every 12 min; ×20 (Nikon CFI Plan Apo Lambda, 0.75 NA) objective lens; 89903-ET491 BV421/BV480/AF488/AF568/AF647 Quinta Band set (Chroma Technology) for multichannel fluorescent images or BV421 only for Hoechst imaging; 16-bit mode with 2 × 2 binning; and 80 ms exposure time at 5% light strength to reduce the light toxicity for Hoechst imaging. Raw images were shading-corrected through NIS-element software to correct for uneven sample illumination, using wells full of imaging medium without cells as the background autofluorescence subtraction.

**Spinning-disk confocal microscopy.** Unless otherwise indicated, imaging was performed using a SoRa spinning-disk confocal microscope (Marianas system, 3i), equipped with a Zeiss Axio Observer 7 stand, ORCA-Fusion BT sCMOS camera (Hamamatsu), CSU-W1 SoRa confocal scanner unit (Yokogawa), and 405, 445, 488, 514, 561 and 637 nm LaserStack (3i). For mTurquoise, mVenus and mCherry 3-colour live-cell imaging, the CSU-W1 dichroic for 445, 515 and 561 nm excitation was

used. Images were acquired every 5 s, 30 s or 2 min based on experiments. The bottom plane was set as the focus plane with the maximal ER–PM contact signals. Definite Focus 2 function was used for long-term focus control. For mTurquoise, mVenus, mCherry and iRFP 4-colour live-cell imaging, the CSU-W1 dichroic for 445, 515 and 561 nm excitation was used for mTurquoise, mVenus and mCherry imaging, whereas the CSU-W1 dichroic for 405, 488, 561 and 640 nm excitation was used for iRFP imaging. For fixed cell imaging, z stack images were captured with a step size of 0.27 µm. Usually, 9 or 13 optical slices were captured based on experiments. For optogenetic experiments, a 3i spinning-disk confocal microscope equipped with an extra 'Vector' photomanipulation device was used. Front regions of migrating cells were manually defined to be illuminated by a 488 nm laser at 1% power every 2 s for continuous activation. Every experiment was pre-imaged for 20 min every 2 min before blue light illumination for another 60 min of imaging.

**Total internal reflection microscopy.** Imaging was performed on a total internal reflection microscope (Nikon) equipped with 488, 561 and 647 nm lasers and high numerical aperture objective at the Rockefeller University's Bio-Imaging Resource Center. Cells stably expressing iRFP–CAAX and eGFP–SEC61β were seeded in a 96-well glass-bottomed plate with pre-patterned linear stripes and fixed for E-Syt1 immunostaining as mentioned above. Signals from the evanescent field were captured and used for quantification.

**Cryo-ET sample preparation and data collection.** Sample preparation and cryo-ET imaging were performed at the New York Structural Biology Center (NYSBC). In brief, 1–2 × 10$^2$ wild-type RPE-1 cells were seeded onto Quantifoil R1/2, 200 mesh gold EM grids with pre-patterned linear stripes prepared as described above. About 12–16 h later after seeding, dishes with grids were taken out of the incubator, and EM grids were blotted at opposite sides of the cells for 1–2 s and plunge frozen with a Leica GP2 (Leica Microsystems). Cryo-patterned grids were visualized using a Titan Krios electron microscope (Thermo Fisher scientific) equipped with a field emission gun, a GIF Quantum LS postcolumn energy filter (Gatan) and a K3 summit electron detector (Gatan). The electron microscope was operated at 300 kv in nanoprobe mode at a magnification of ×19,500 (pixel size of 4.53 Å at the specimen level). Cryo-ET tilt-series were collected using a dose symmetric scheme[49] with a tilt range of −52° to +52° at a target defocus of −8 µm and 3° increments in SerialEM for a total dose of 126 e Å$^{-2}$.

### 3D collagen gel migration
PureCol (Advanced Biomatrix, 5005-B, 3 mg ml$^{-1}$) was used for 3D collagen gel migration assays. Before gelation, all collagen-related steps were performed on ice. Eight parts of collagen solution was slowly mixed with 1 part of chilled 10× PBS to prepare 2.4 mg ml$^{-1}$ collagen stock solution, the pH of which was adjusted to 7.0–7.5 using sterile 0.1 M NaOH. Next, 60 µl of 0.5 mg ml$^{-1}$ collagen solution (stock solution was diluted into cold 1× PBS) was added to a 96-well plate and left for 2 h at 37 °C for polymerization. The gel was washed with PBS for 3 times, and 50 µl of RPE-1 cells at a density of 2 × 10$^4$ cells per ml was added into each well for 2–3 h attachment. After incubation, 40 µl of medium was removed from the well and 60 µl of collagen was added. The 96-well plate with collagen on top of the cells was incubated at 37 °C for another 2–3 h and washed again with complete growth medium 6 times and cultured in a 37 °C incubator for 24 h before imaging.

### Quantification and image analysis
Automated analysis of time-lapse imaging was performed using a custom Matlab R2021 pipeline based on previous work[9,50]. Details and parameters regarding how to quantify signals are summarized as below.

**Cell segmentation and time-lapse tracking.** Cells were automatically segmented from either the nuclear signal or the PM signal based on

experiments. For 2D cell migration assays, nuclear signals (Hoechst staining) captured using an automated epifluorescence microscope were used for segmentation based on Laplacian of Gaussian algorithms. The detected nuclei were tracked using a nearest-neighbour algorithm between the current frame and its previous frame. To increase tracking accuracy, the nuclear mass (the product of nuclear intensity and nuclear area) was also used as a constant metric to adjust matching between two neighbouring frames. After tracking, $x$ and $y$ coordinates of each nucleus were exported for cell speed and MSD calculation. Cell speed was quantified as the value of total travelling distance over travelling time. MSD was quantified as the mean value of $(xy(t) - xy(0))^2$, where $xy(t)$ is the position of cell at time point $t$ and $xy(0)$ is the initial position. For 1D cell migration assays, the PM signal (CAAX fluorescence intensity), captured by spinning disk confocal microscope, was used for segmentation based on a modified version of Otsu's algorithm. With the help of linear stripes, only isolated single cells were selected for subsequent analysis. Nearest-neighbour algorithm between subsequent frames was used for cell tracking. Cell velocity was determined based on the overall centroid distance travelled over a fixed time. A first-order polynomial function was used to fit cell trajectory based on $xy$ coordinates to determine cell direction, which was smoothed to minimize the effects of random centroid movements.

**Time lapse of MAPPER puncta tracking.** To track ER–PM contacts, the MAPPER reporter signal was captured every 20 s and subjected to tophat filtering (3 pixel radius), followed by cell segmentation based on the CAAX signal as discussed above. The threshold for puncta segmentation was determined by the MAPPER signals located at 40% from the cell front using a modified version of Otsu's algorithm. This facilitated identification of ER–PM contacts at the cell front, which were much dimmer than those at the back. A nearest-neighbour algorithm was used to connect puncta between subsequent frames, and the MAPPER mass was set as a constant variable to adjust mismatched puncta. The speed of puncta was determined by their travelling distance over duration. The lifetimes of MAPPER puncta were calculated as the duration from its appearance to disappearance or the last frame. The MAPPER mass was quantified as the product of MAPPER intensity and MAPPER area.

**Kymograph plots and gradient profiles in 1D migrating cells.** A schematic graph of the kymograph and average gradient analysis are shown in Extended Data Fig. 1d. In brief, the CAAX signal at each time point was used to mask individual cells and a 3 μm ring around the cell periphery. The latter was used to include a region with higher relative PM contribution. To minimize the effect of cell shape differences among cells, each cell mask and ring mask were divided into 20 equal segments from the cell front to the back. The cell front was automatically identified as the tip end, which has the farthest distance from the centroid of the cell mask. Each pixel value of the normalized biosensor intensity, biosensor size or biosensor mass in the cell and ring mask were assigned to 1 of the 20 segments. An averaged value in each segment from the front to back was calculated to create profiles at each time point that were then represented in a kymograph. The average of the gradient profiles at the different time points or from different cells is also shown in most figures. Unless otherwise stated, the time series of kymograph profiles is shown as a heatmap, with the $x$ axis representing time and each vertical line representing the 20 bins of the spatial distribution of a specific parameter at a specific time. For the kymograph profiles of the MAPPER sensor, the full cell mask was used, whereas the ring mask was used for the pan-Tyr signal.

**Growth rate analysis and slope calculation.** For the MAPPER growth rate analysis (Fig. 4h), the CAAX signal was used to define the cell boundary and a region of interest (ROI) was chosen near the cell front as indicated (Extended Data Fig. 5f). As the cell migrated, the ROI moved and became increasingly located towards the back. The MAPPER to CAAX ratio of each MAPPER puncta in the ROI was quantified and plotted versus the ROI position. To compare the growth rate differences in the front versus back in Fig. 4i, the averaged curve of growth rate was split into the front part and the back part, which were separately fitted by a linear regression function to calculate the slope. The front segment was defined as the relation position from 0.2 to 0.5, whereas the back part was from 0.5 to the position where the MAPPER puncta showed the maximal average intensity.

**Polarity steepness.** In Figs. 2d and 4f and Extended Data Fig. 5b,e, polarity steepness was calculated from normalized MAPPER to CAAX or MAPPER to SEC61β profiles in migrating cells. The average polarity score in Fig. 2d was quantified by normalizing the absolute value of (mean (front 20%) − mean (back 20%)) to the mean (back 20%). In Fig. 4f and Extended Data Fig. 5b,e the MAPPER signals between the back and front 50% of cell were measured and the ratio of front to back was calculated to represent the polarity steepness.

**Cryo-ET data processing.** The raw tilt movies were motion-corrected and CTF-corrected in Warp (v.1.09)[51]. The tilt-series stack was exported from Warp, in which the tilt series were aligned with the AreTomo software package[52]. The aligned tilt series were reconstructed with either weighted-back projection in AreTomo or with using Tomo3D[53]. To enhance the contrast of the tomograms for visualization, weighted-back projected and simultaneous iterative reconstructive technique tomograms were CTF-deconvolved with IsoNet[54]. Initial segmentations were manually performed on a few tomographic slices in DragonFly (v.2022.2) for training a neural network in DragonFly (v.2022.2)[55] or by using Tardis-Pytorch[56] on the deconvolved tomograms. Segmentations were corrected and manually labelled using DragonFly. The segmentations were subsequently processed and analysed using the surface morphometrics analysis toolkit[57]. Curvedness was calculated for the front and back ER using the surface morphometrics analysis toolkit and was chosen as it is an unassigned combination of the two principal components of curvature and is used because the surface normal vectors do not have a sign.

## Statistical analysis

Statistical results were analysed using GraphPad Prism 8.0 or Matlab R2021 and shown as the mean ± s.d. or mean ± s.e.m. as indicated. Comparisons were made between groups using unpaired two-tailed Student's $t$-test or one-way ANOVA and Scheffe's/Dunnett post hoc comparison as indicated. For the pTyr to CAAX or the MAPPER to CAAX gradient profile significance test, 20% front and back ratio was used in Figs. 1b,c and 2f,i and Extended Data Fig. 8a,d,g, whereas the index of peak ratio was used in Figs. 3c,d and 5i and Extended Data Fig. 7f,h. For all analyses, $*P < 0.05$, $**P < 0.01$ and $***P < 0.001$ were considered significant. NS indicates statistical non-significance with $P > 0.05$. Each experiment was performed at least three independent times.

## Reporting summary

Further information on research design is available in the Nature Portfolio Reporting Summary linked to this article.

## Data availability

The authors declare that all data supporting the findings of this study are available within the article and its supporting information files (extended data and supplementary videos). Cryo-ET raw data are available from FigShare (https://doi.org/10.6084/m9.figshare.25621119)[58]. Source data are provided with this paper.

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

**Acknowledgements** We thank all members of the T.M. Laboratory and M. Teruel Laboratory for discussions and sharing reagents; staff at the flow cytometry core facility for cell sorting; and A. Hayer and D. Rosenthal for their comments on this manuscript. The work was supported by a grant from the National Institute of General Medical Sciences (NIGMS) (R35GM127026 to T.M.). Cryo-ET experiments were supported by the Simons Electron Microscopy Center and the National Resource for Automated Molecular Microscopy located at the New York Structural Biology Center, supported by grants from the Simons Foundation (SF349247) and the NIH National Institute of General Medical Sciences (GM103310).

**Author contributions** Investigation of ER–PM contact sites was initially proposed by B.G. and confirmed by T.M., and the research plan and the conceptualization of the results were based on discussions between B.G. and T.M. B.G. and T.M. wrote the manuscript. B.G. performed most of the experiments and analyses. J.D.J. imaged the cryo-ET grids made by B.G. and did the initial segmentation and curvature analysis of cryo-ET images with the help of B.G. and A.d.M. A.T. helped with the optimization of the code for gradient analysis.

**Competing interests** The authors declare no competing interests.

**Additional information**
**Correspondence and requests for materials** should be addressed to Bo Gong or Tobias Meyer.

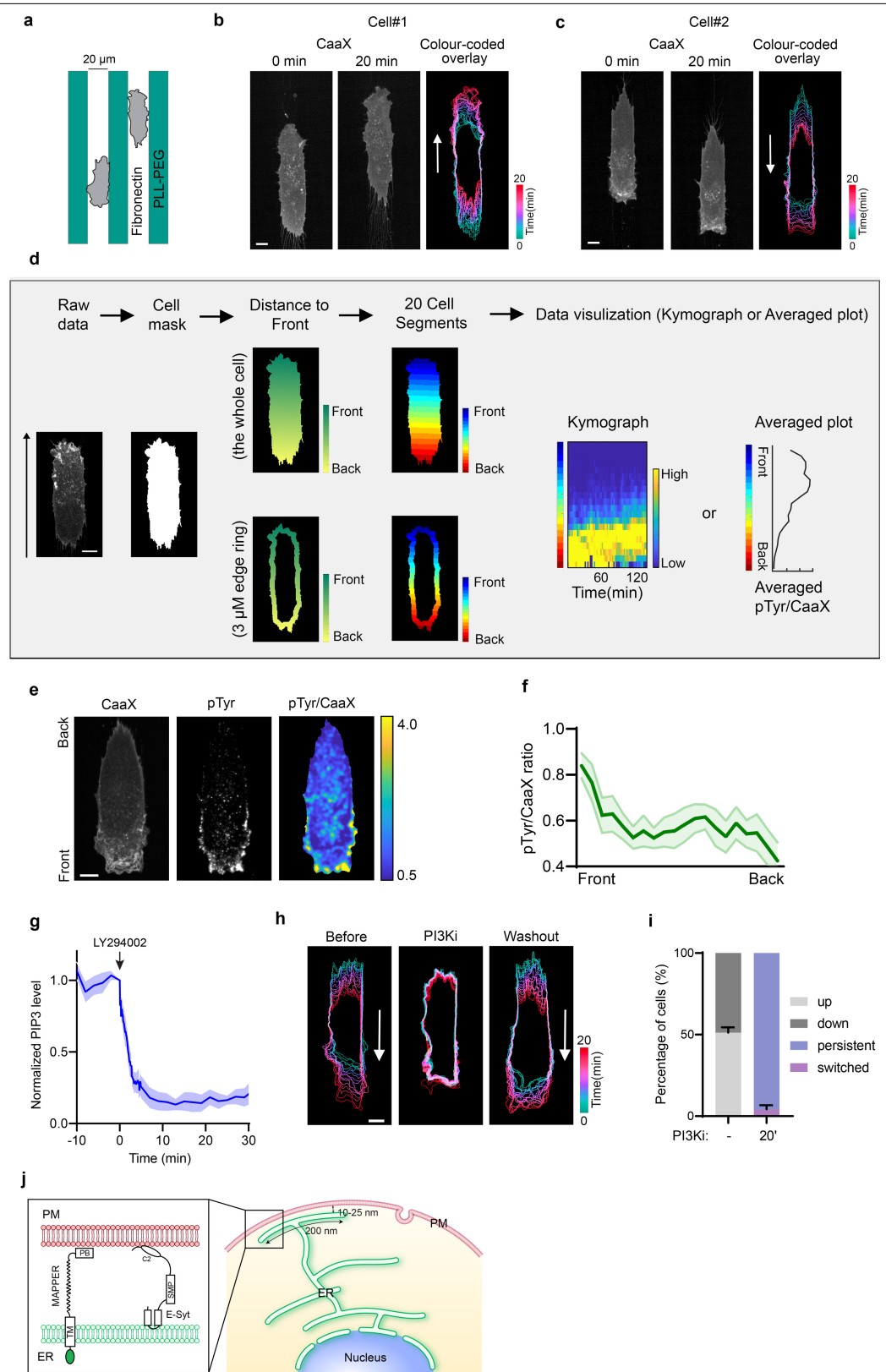

**Extended Data Fig. 1** | See next page for caption.

**Extended Data Fig. 1 | RPE-1 cells on linear stripes show persistent movement, polarized pTyr signaling. a**, Schematic of micropatterned linear fibronectin stripes. **b**, **c**, Representative examples of two cells, moving randomly up (**b**) or down (**c**) on linear stripes upon uniform stimulation. Left and middle panels show mTurquoise-CaaX images taken 20 min apart. The mTurquoise-CaaX signal was used to detect the cell boundaries in sequential images. Right, cell boundaries were derived from images taken at 2-minute intervals over 20 min, shown color-coded from cyan to red. Migration directions are shown by arrows. **d**, Schematic of the kymograph and average front-to-back gradient analysis used in the study. For the kymograph and gradient analysis, the mTurquoise-CaaX signal from individual RPE1 cells is used to create the whole cell mask and a 3-μm ring mask close to cell periphery, respectively. Based on the distance to the front, the cell and ring mask are binned into 20 equally distributed segments and the biosensor activity is visualized either as a heatmap kymograph or averaged gradient plot, respectively. **e**, pan-pTyr staining of RPE-1 cell expressing a mTurquoise-CaaX PM marker migrating on stripe. Right panel, pTyr/CaaX ratio image. Color scale marks ratio intensity (typical example out of n = 17 cells analyzed). **f**, Averaged plot of pTyr/CaaX ratio of RPE1 cells migrating on linear stripes. RPE1 cells were seeded on linear stripes and fixed for pan-Tyr staining. n = 17 cells. **g**, Profiles of Akt1-PH-mCherry PM intensity before and after addition of PI3K inhibitor LY294002. RPE1 cells stably expressing mTurquoise-CaaX, MAPPER-mVenus and Akt1-PH-mCherry were seeded on linear stripes for time-lapse imaging. Prior to 10 μM LY294002 treatment, cells are captured every 2 min for 10 min. Upon LY294002 addition, images are captured every 30 s for 5 min and then captured every 2 min for another 25 min. n = 5 cells. **h**, Color-coded cell boundaries before, during, and after a 20-minute application of 10 μM LY294002 (PI3Ki) (2 min interval). Arrow marks the migration direction (typical example out of n = 913 cells analyzed). **i**, Fraction of cells that migrate in the same direction before and after stalling for 20 min in response to addition and removal of 10 μM LY294002. Cells were captured every 5 min for 30 min and treated with 10 μM LY294002 treatment for 20 min. After removing the inhibitor, cells were captured every 5 min for another 60 min. Cell direction is determined by fitting cell trajectory with a first order polynomial function. Persistent movement was defined as a cell having the same direction in the first 30 min (before inhibitor treatment) and last 60 min after the washout (after inhibitor washout). A switched movement means a cell that reverses the direction. n = 913 cells analyzed for the PI3Ki group. Control analysis showing a random choice of the cell migration direction for n = 858 cells plated on linear stripes (up and down grey bars as in Extended Data Fig. 1b and c). Mean ± s.d. **j**, Schematic of ER-PM contact sites formed by E-Syt tether proteins and marked by MAPPER. Mean (dark) ± s.e.m. (shaded) in (**f**, **g**).

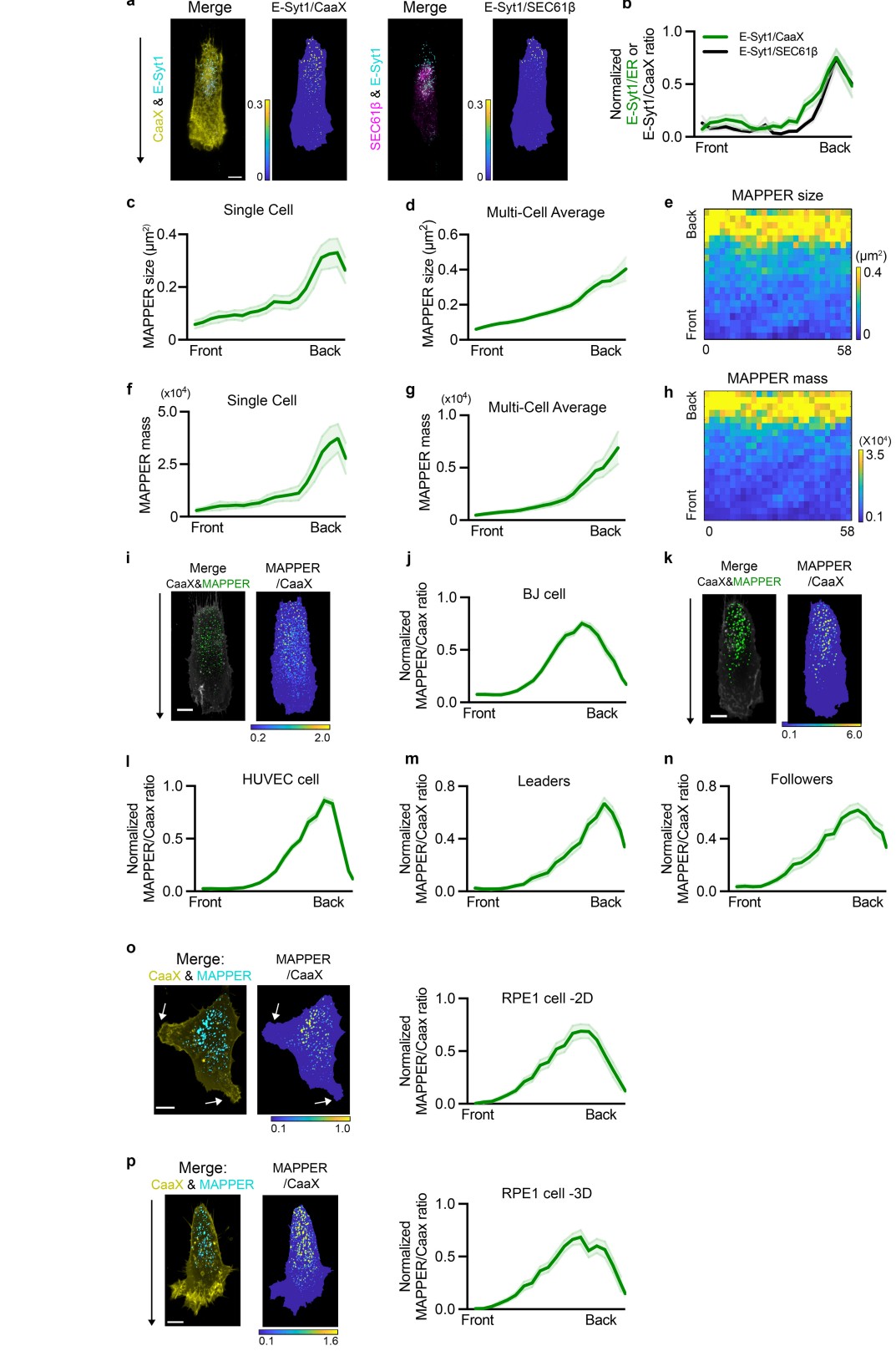

**Extended Data Fig. 2 |** See next page for caption.

**Extended Data Fig. 2 | Migrating cells on linear stripes show ER-PM contact gradient. a**, Representative TIRF image of fixed RPE-1 cells stably expressing iRFP-CaaX (yellow) and EGFP-SEC61β (magenta) after E-Syt1 staining (cyan). Right, E-Syt1/CaaX and E-Syt1/Sec61β ratio color images. Scale of E-Syt1/CaaX and E-Syt1/Sec61β is normalized for maximal signal range as shown in corresponding color bars. **b**, Mean E-Syt1/CaaX and E-Syt1/Sec61β gradient. n = 14 cells. **c**, Front-to-back gradient of the MAPPER puncta size of cell in Fig. 1e; time averaged gradient (green). **d**, Group averaged gradients of the MAPPER size (n = 23 cells). **e**, Kymograph of ER-PM contact area size over time, aligned by their relative front-to-back location. Time interval, 2 min. **f-h**, Same analysis as **c-e** for the MAPPER puncta mass, the product of MAPPER intensity and area. **i**, Representative BJ-5ta fibroblast cells stably expressing mTurquoise-CaaX and MAPPER-Venus on stripe. Right, MAPPER/CaaX ratio image. Scale of MAPPER/CaaX is normalized for maximal signal range as shown in corresponding color bars. Migration direction is indicated by arrow. **j**, Mean MAPPER/CaaX gradients in BJ-5ta cells (n = 57 cells). **k,l**, Same for HUVEC endothelial cell. n = 35 cells. **m,n**, Mean MAPPER/CaaX gradients in leader cells (**m**, n = 46 cells) and follower cells (**n**, n = 33 cells) during collective cell migration. **o**, RPE-1 cells migrating on collagen coated glass in 2D. White arrows denote 2 membrane protrusions. Middle, maximal projection MAPPER/CaaX ratio images. Right, averaged MAPPER/CaaX ratio of 19 cells. **p**, Same for RPE-1 cells migrating in 3D within 0.5 mg/ml collagen matrix. n = 17 cells. Scale bar, 10 μm. Mean (dark) ± s.e.m. (shaded) in (**b**, **c**, **d**, **f**, **g**, **j**, **l-p**).

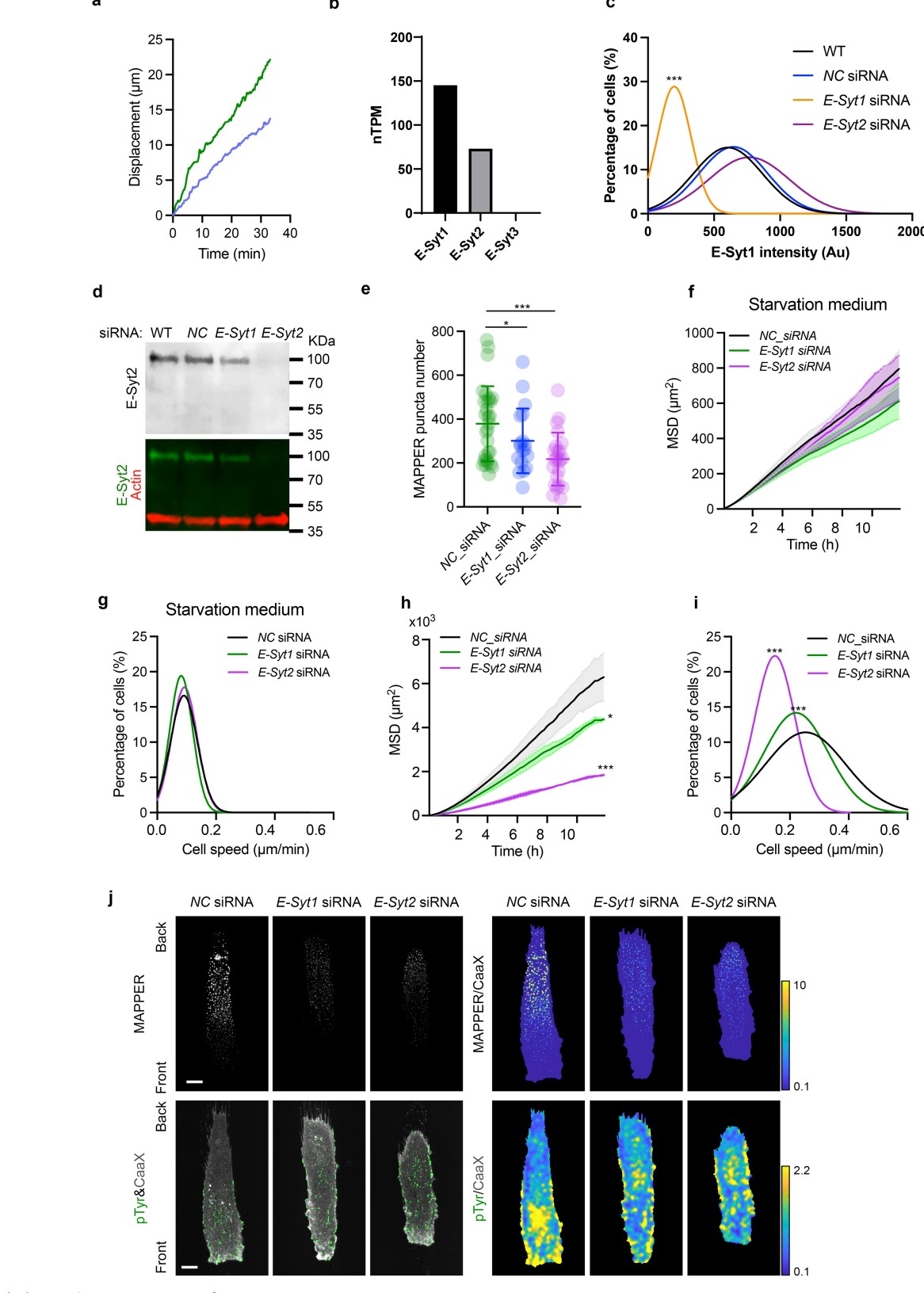

**Extended Data Fig. 3** | See next page for caption.

**Extended Data Fig. 3 | E-Syt1 or E-Syt2 knockdown in RPE-1 cells reduces the number of ER-PM contact sites and suppresses cell migration. a**, Centroid displacement graph for the 2 marked example cells in (Fig. 2d). **b**, Relative transcript expression of *E-Syt1*, *E-Syt2* and *E-Syt3* in RPE1 cells. **c**, Single-cell immunofluorescence analysis of wild type cells and cells with indicated siRNA transfection. Wild type and siRNA-transfected cells were fixed for E-Syt1 immunostaining. n = 32048, 25369, 28396 and 25349 cells for wild type cells, control, *E-Syt1* siRNA and *E-Syt2* siRNA group. Control vs. *E-Syt1* siRNA, p-value = 0. **d**, Western-blot analysis of cell lysates from WT, control and target siRNA transfected cells. E-Syt2 and Actin were detected. For gel source data, see Supplementary Fig. 1a. **e**, Number of ER-PM contacts in RPE1 cells transfected with control, *E-Syt1* or *E-Syt2* siRNA. Cells were fixed after 48 h transfection for MAPPER puncta number quantification using automated segmentation. n = 29, 18 and 23 cells for control, *E-Syt1* and *E-Syt2* siRNA group. Control: p-value = (vs. *E-Syt1* siRNA) 0.0162 and (vs. *E-Syt2* siRNA) 0.0006. **f**, Mean square displacement of RPE-1 cells transfected with control, *E-Syt1* or *E-Syt2* siRNAs in starvation medium on 2D surfaces over 12 h. Cells transfected with siRNAs were starved overnight and imaged for another 12 h. n = 4665, 4808 and 4149 cells for control siRNA, *E-Syt1* siRNA, *E-Syt2* siRNA group respectively. **g**, Quantification of mean cell speed for the RPE-1 cells in (**f**). Frequency distribution of mean cell speed for cells are shown. Goodness of fit, $R^2$ = 0.9599, 0.9760 and 0.9713 for control siRNA, *E-Syt1* siRNA and *E-Syt2* siRNA group respectively. **h**, **i**, Same as for f and g but with growth instead of starvation medium. n = 1358, 1921 and 1387 cells for control siRNA, *E-Syt1* siRNA and *E-Syt2* siRNA respectively. Control: p-value = (vs. *E-Syt1* siRNA) 0.0327 and (vs. *E-Syt2* siRNA) 7.0842×10$^{-5}$. Goodness of fit in (**i**), $R^2$ = 0.9453, 0.9619 and 0.9644 for control siRNA, *E-Syt1* siRNA and *E-Syt2* siRNA group respectively. Control: p-value = (vs. *E-Syt1* siRNA) 1.5120×10$^{-13}$ and (vs. *E-Syt2* siRNA) 2.1619×10$^{-87}$. **j**, Representative MAPPER/CaaX and pTyr/CaaX ratio images of RPE-1 cells transfected with control, *E-Syt1* or *E-Syt2* siRNAs. RPE-1 cells stably expressing mTurquoise-CaaX (gray) and MAPPER-mVenus (magenta) were seeded on linear stripes and transfected with control or *E-Syt* siRNAs. 48 h after transfection, cells were fixed for pan-Tyr antibody staining. Scale of MAPPER/CaaX is normalized for maximal signal range. Scale bar is 10 μm. One-way ANOVA and Scheffe's post hoc comparison in (**c**, **e**, **h**, **i**). mean ± s.d. in (**e**, **f**, **h**).

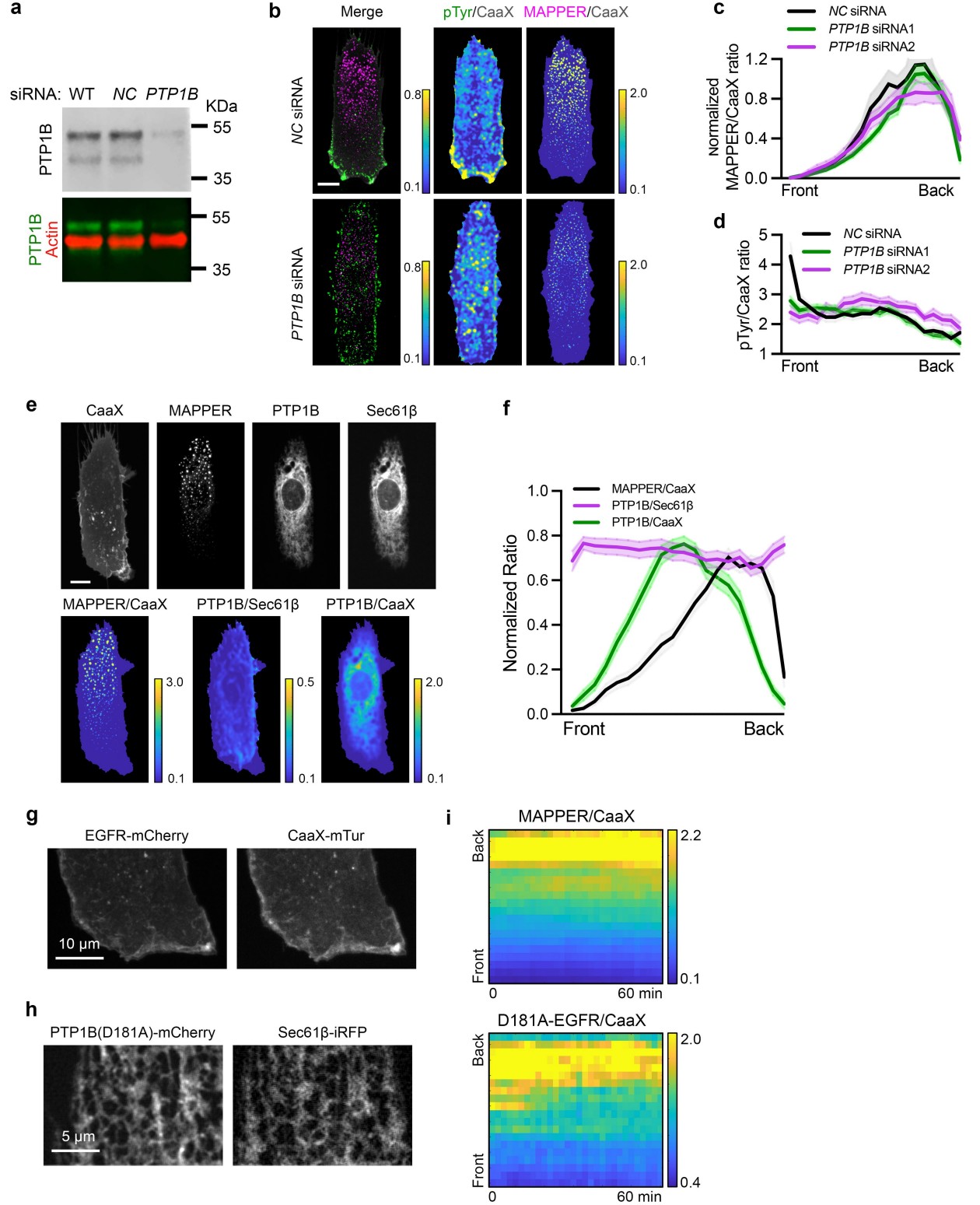

**Extended Data Fig. 4 | PTP1B is evenly distributed in migrating RPE-1 cells and knockdown of PTP1B disrupts the polarized pan-Tyr gradient.** **a**, Western-blot analysis of cell lysates from WT, control and *PTP1B* siRNA transfected cells. PTP1B and Actin were detected. For gel source data, see Supplementary Fig. 1b. **b**, Representative MAPPER/CaaX and pTyr/CaaX ratio images of RPE-1 cells transfected with control versus *PTP1B* siRNA. RPE-1 cells stably expressing mTurquoise-CaaX (gray) and MAPPER-mVenus (magenta) seeded on linear stripes. Cells were fixed and stained using a pan-Tyr antibody (green). **c**,**d**, Average gradient of MAPPER/CaaX ratio (**c**) or pan-Tyr/CaaX ratio (**d**) in RPE-1 cells, comparing control and transfection with two *PTP1B* siRNAs.

n = 19, 36 and 26 cells for control siRNA (black), *PTP1B* siRNA1 group (green) and *PTP1B* siRNA2 group (magenta) respectively. **e**, Representative PTP1B/CaaX and PTP1B/SEC61β ratio images of RPE-1 cells stably expressing mTurquoise-CaaX, MAPPER-mVenus, PTP1B-mCherry and SEC61β-iRFP. **f**, Averaged normalized PTP1B/CaaX and PTP1B/SEC61β gradients using images as in (**e**). n = 31 cells. **g**, Subcellular localization of EGFR-mCherry protein in RPE-1 cells with mTurquoise-CaaX expression. Scale bar, 10 μm. **h**, Subcellular localization of PTP1B(D181A)-mCherry protein in RPE-1 cells with SEC61β-iRFP expression. Scale bar is 5 μm. **i**, Example kymographs of MAPPER/CaaX and PTP1B(D181A)-EGFR/CaaX ratio changes. Mean (dark) ± s.e.m (shaded) in (**c**, **d**, **f**).

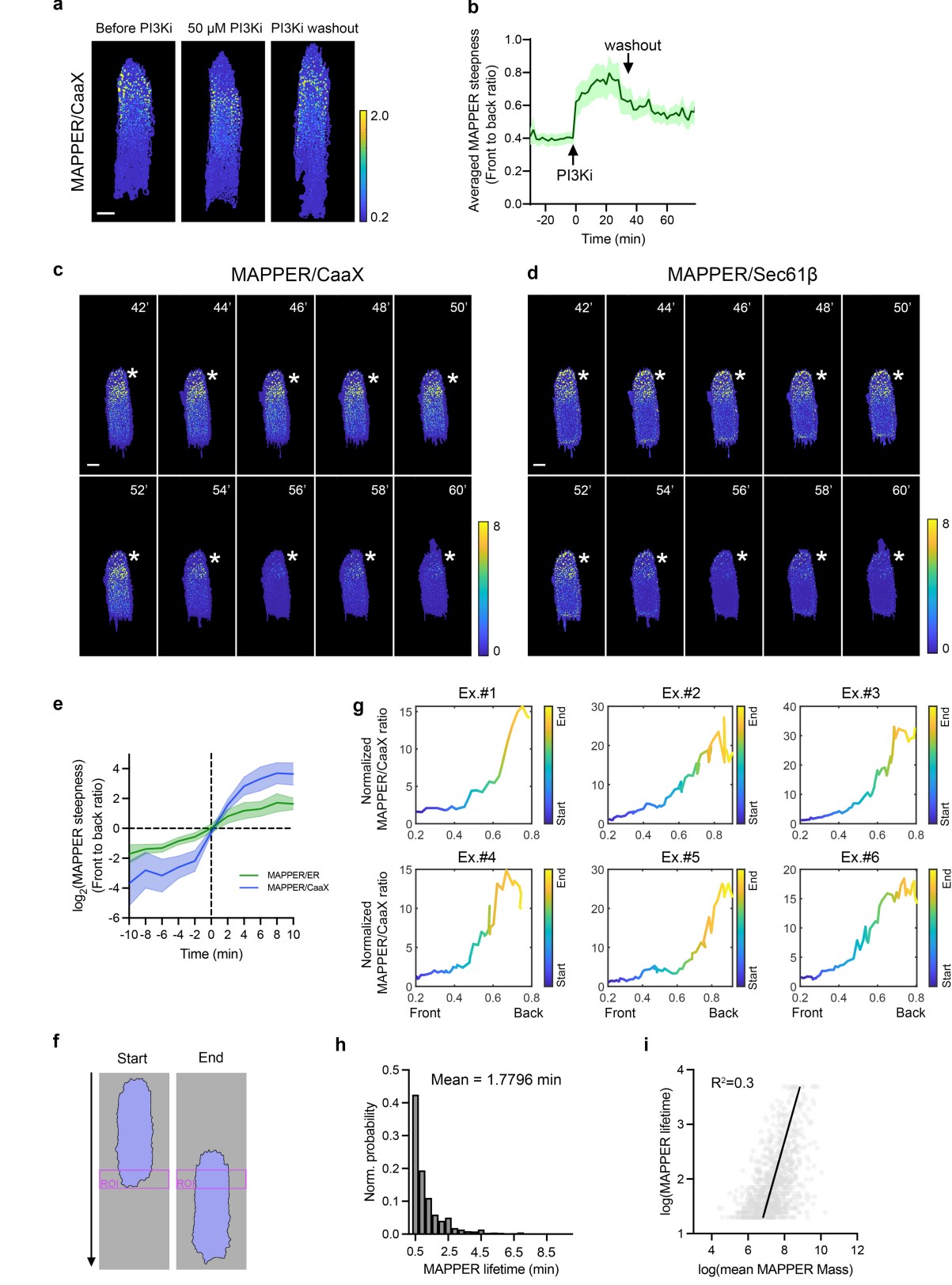

**Extended Data Fig. 5 |** See next page for caption.

**Extended Data Fig. 5 | ER-PM contacts show a higher growth rate in the back and cells maintain their back-to-front gradient even after they stop moving.** **a**, Representative MAPPER/CaaX ratio images in RPE-1 cells before, 10 min after 10 μM LY294002 treatment and 20 min after LY294002 washout. RPE1 cells stably expressing mTurquoise-CaaX and MAPPER-mVenus were seeded on linear stripes for time-lapse imaging. Prior to 10 μM LY294002 treatment, cells are captured every 2 min for 30 min before LY294002 was added to the medium. After 20 min inhibition, LY294002 was washed out and cells were imaged for another 30 min. **b**, Averaged profiles of MAPPER steepness of RPE-1 cells treated as in (**a**). The MAPPER steepness is represented by the ratio of the front half to back half MAPPER/CaaX ratio and aligned to the time point when the PI3K inhibitor was added in. All ratios under 1 are polarized MAPPER signal towards the back. The black arrows mark the time points when the inhibitor was added and when it was washed out. n = 7 cells. **c**, **d**, Representative MAPPER/CaaX (**c**) and MAPPER/SEC61β (**d**) ratio images of RPE-1 cells migrating towards the end of a linear stripe and reversing the direction of migration. RPE-1 cells stably expressing mTurquoise-CaaX, MAPPER-mVenus and SEC61β-iRFP. The ratio images are shown at indicated time points. Time interval is 2 min. Scale bar, 10 μm. **e**, Change in MAPPER gradient 10 min before and after a turn. MAPPER steepness in log2 scale. n = 53 cells. **f**, Schematic of a stationary region of interest (ROI) marking the front and back at different times for growth rate analysis in Fig. 4h (see methods). **g**, Representative examples of growth rate curves from 6 different ER-PM contact sites. Six ER-PM contacts in the ROI are taken as examples to show their growth rate, represented by their MAPPER/CaaX ratios. These ratios are aligned to the ROI position relative to the migrating cell and the growth curve is also pseudo color-coded with the time. **h**, Histogram of MAPPER puncta lifetime. 60238 MAPPER puncta from 6 individual RPE-1 cell migrating on stripes were tracked and the distribution of puncta lifetime is shown. **i**, Scatter plot of MAPPER puncta mass versus MAPPER lifetime with fitted regression line. MAPPER mass is the product of MAPPER intensity and area (n = 993 puncta, $R^2$ = 0.3). Mean (dark) ± s.e.m. (shaded) in (**b**, **e**).

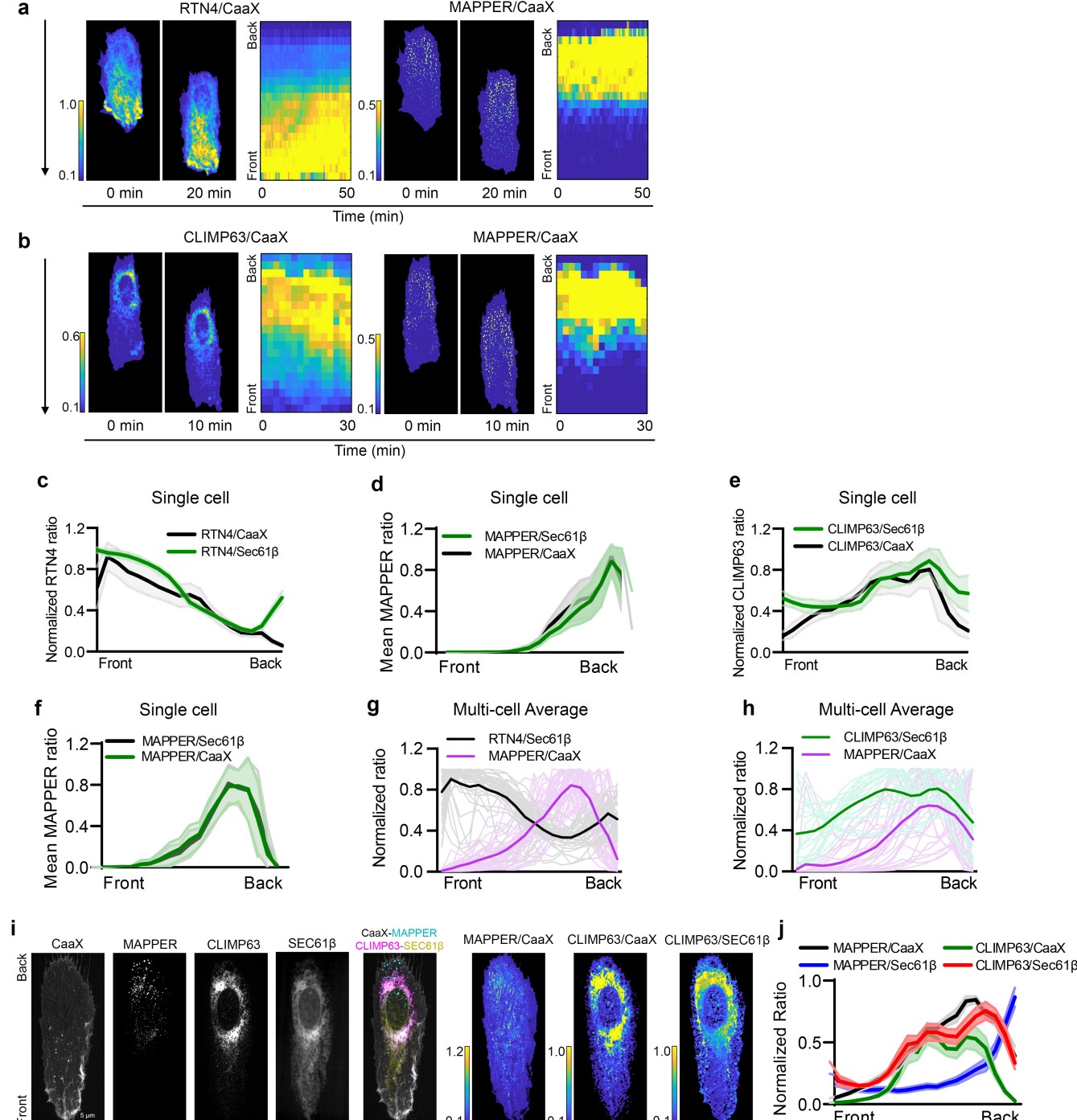

**Extended Data Fig. 6 | The curvature of the ER in migrating RPE-1 cells is polarized, with curved ER in the front versus flatten ER in the back.**
**a**, **b**, RTN4/CaaX (**a**) or CLIMP63/CaaX (**b**) and MAPPER/CaaX ratio images at indicated time points. Kymograph plots over time from Fig. 5b,c.
**c**,**d**, Quantification of RTN4 (**c**) or MAPPER (**d**) gradient changes over 50 min for the cell in Fig. 5b. Black lines indicate normalization to the CaaX signal and green lines are normalization to the SEC61β signal. **e**,**f**, Quantification of CLIMP63 (**e**) or MAPPER (**f**) ratio changes over 30 min for the cell in Fig. 5c. **g**, Close match of the RTN4/SEC61β ratio profile with the inverted MAPPER/CaaX profile. The black line represents the averaged RTN4/SEC61β ratio profile from 37 cells, indicated by every single light gray line. The magenta line represents the averaged MAPPER/CaaX ratio profile from 37 cells, indicated by every single

light magenta line. **h**, Correlation of CLIMP63/SEC61β ratio profile with MAPPER/CaaX ratio. The green line represents the averaged CLIMP63/SEC61β ratio profile from 23 cells. Multiple light green lines mark each cell. The magenta line represents the averaged MAPPER/CaaX ratio profile from 23 cells, Light mangenta lines mark individual cells. **i**, Endogenous CLIMP63 distribution in RPE-1 cells stably expressing mTurquoise-CaaX, MAPPER-mVenus and SEC61β-iRFP on linear stripe. RPE-1 were seeded on linear stripes and fixed and stained with CLIMP63 antibody. MAPPER/CaaX, CLIMP63/CaaX and CLIMP63/SEC61β ratio images are shown with scaled colors as indicated. **j**, Averaged profiles of MAPPER/CaaX (black), MAPPER/SEC61β (blue), CLIMP63/CaaX (green) and CLIMP63/SEC61β ratio (red). n = 44 cells. Mean (dark) ± s.d. (shaded) in (**c**-**f**,**j**).

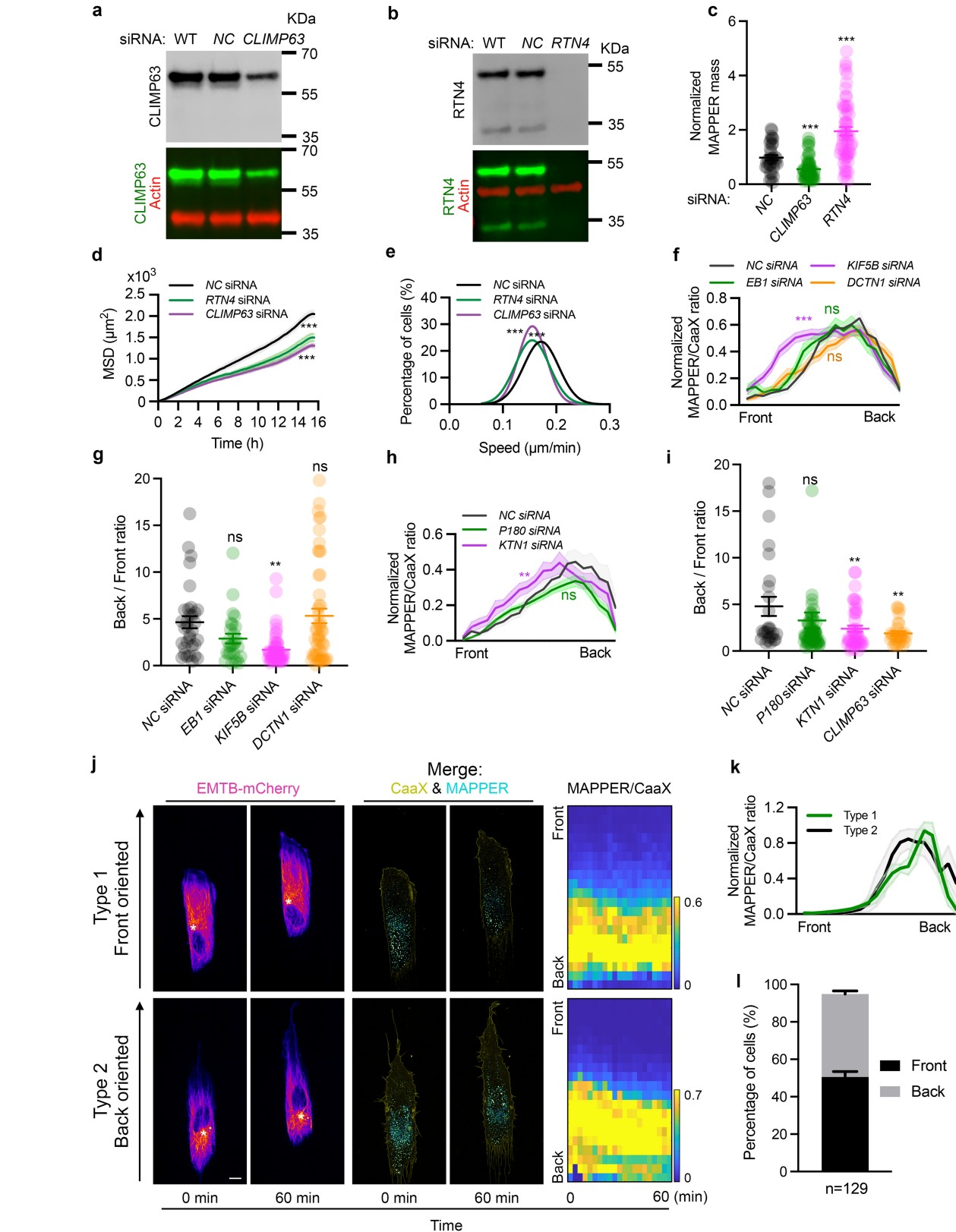

**Extended Data Fig. 7 |** See next page for caption.

**Extended Data Fig. 7 | Knockdown of ER-shaping proteins, ER-MT connection related proteins. a,b,** Western-blot analysis of cell lysates from WT, control and *CLIMP63* (**a**) or *RTN4* (**b**) siRNA transfected cells. CLIMP63 (**a**) or RTN4 (**b**) and Actin were detected. For gel source data, see Supplementary Fig. 1c,d. **c,** MAPPER mass in cells treated with control, *RTN4* or *CLIMP63* siRNAs. n = 8947 puncta from 34 control cells, 13952 puncta from 56 *RTN4* siRNA transfected cells and 38946 puncta from 58 *CLIMP63* siRNA transfected cells. Control: p-value = (vs. *CLIMP63* siRNA) $4.0558 \times 10^{-99}$ and (vs. *RTN4* siRNA) $4.0409 \times 10^{-120}$. **d, e,** Mean square displacement (**d**) and averaged speed distribution (**e**) of RPE-1 cells transfected with control, *RTN4* or *CLIMP63* siRNAs when stimulated by 20 ng μl⁻¹ EGF on 2D surfaces over 16 h, after overnight starvation before EGF stimulation. n = 1831, 1741, 2513 cells for control, *CLIMP63* siRNA and *RTN4* siRNA respectively, Control: p-value = (vs. *CLIMP63* siRNA) $4.8248 \times 10^{-13}$ and (vs. *RTN4* siRNA) $2.0467 \times 10^{-08}$ in (**d**). n = 2368, 2281 and 1436 cells for control, *CLIMP63* siRNA and *RTN4* siRNA respectively. Control: p-value = (vs. *CLIMP63* siRNA) $1.3114 \times 10^{-39}$ and (vs. *RTN4* siRNA) $1.6008 \times 10^{-24}$ in (**e**). **f,** Quantification of MAPPER/CaaX profile of RPE-1 cells transfected with control, *KIF5B, EB1* or *DCTN1* siRNAs. n = 46, 64, 27 and 66 for control, *KIF5B* siRNA, *EB1* siRNA and *DCTN1* siRNA group respectively. Control: p-value = (vs.

*KIF5B* siRNA) 0.0005, (vs. *EB1* siRNA) 0.076 and (vs. *DCTN1* siRNA) 0.4198. **g,** same as (**f**) but back to front ratio of MAPPER/CaaX ratio was analyzed. Control: p-value = (vs. *KIF5B* siRNA) 0.0065, (vs. *EB1* siRNA) 0.3663 and (vs. *DCTN1* siRNA) 0.8837. **h,** same as (**f**) but cells transfected with control, *P180* or *KTN1* siRNAs. n = 26, 73 and 53 cells for control siRNA, *P180* siRNA and *KTN1* siRNA. Control: p-value = (vs. *P180* siRNA) 0.1025 and (vs. *KTN1* siRNA) 0.0069. **i,** same as (**h**) but back to front ratio of MAPPER/CaaX ratio was analyzed. n = 29, *CLIMP63* siRNA. Control: p-value = (vs. *P180* siRNA) 0.3272, (vs. *KTN1* siRNA) 0.0092, (vs. *CLIMP63* siRNA) 0.0062. Unpaired two-tailed t test. **j,** (left) Pairs of images captured 60 min apart of representative RPE-1 cells stably expressing MAPPER-mVenus (cyan), along with mTurquoise-CaaX (yellow) and EMTB-mCherry (magenta). Right, kymographs of the average back-to-front intensity ratio over time (time interval is 2 min). Positions of MTOC are indicated by the white stars. **k,** Averaged gradients of MAPPER/CaaX ratio from the cell in (**j**). **l,** Percentage of cells with front versus back orientated MTOC position. n = 129 cells from 3 independent experiments. One-way ANOVA and Scheffe's post hoc comparison in (**c** – **h**). Mean (dark) ± s.e.m. (shaded) in (**c, f, g-i, k**) and Mean (dark) ± s.d. (shaded) in (**d, l**).

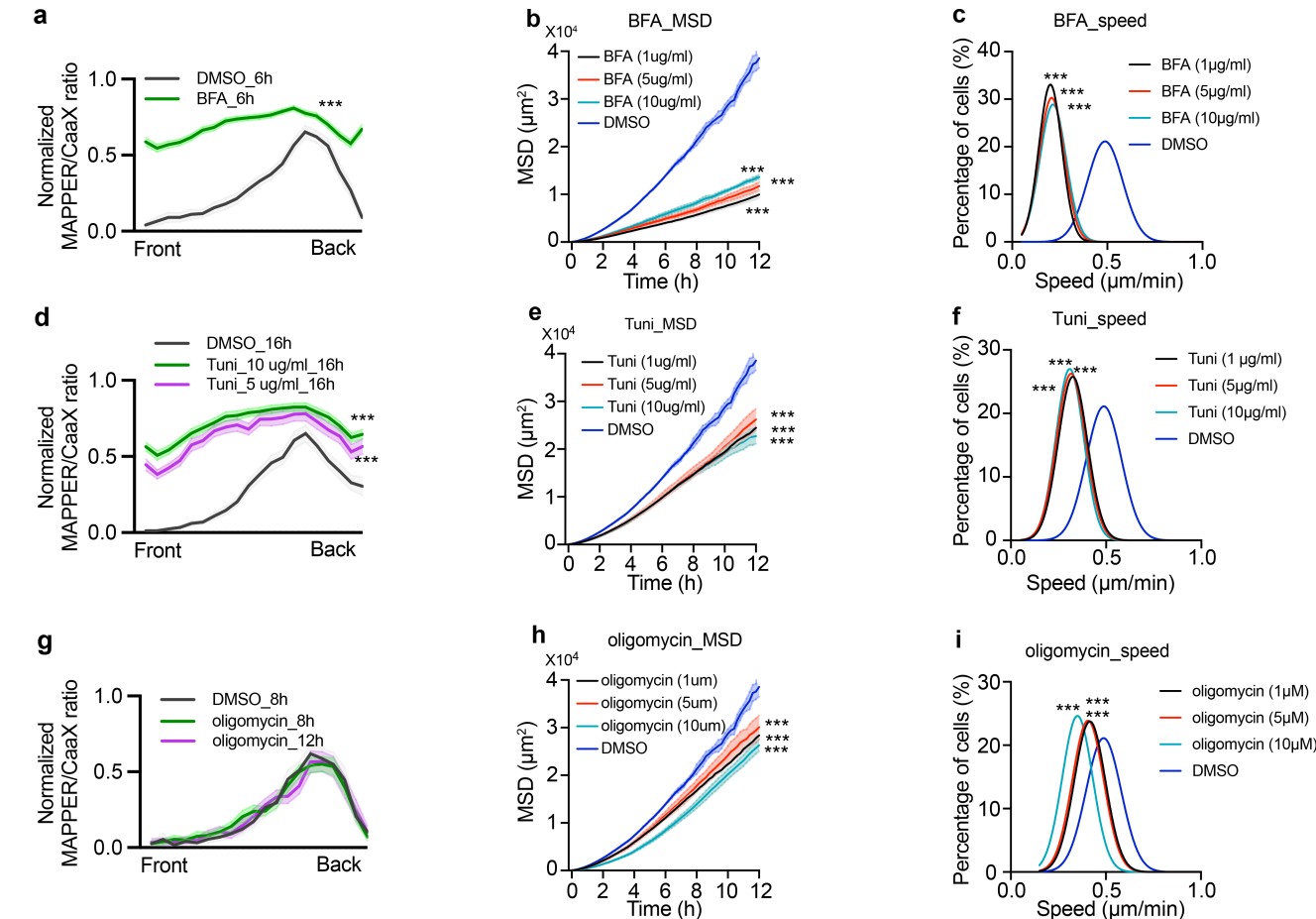

**Extended Data Fig. 8 | Blockage of ER homeostasis in RPE-1 cells reduces ER-PM contact gradients and suppresses cell migration. a-c**, Mean gradients of MAPPER/CaaX ratio (**a**), mean square displacement (**b**) and mean cell speed (**c**) analysis of RPE-1 after 6 h DMSO or BFA (10 μg/ml) treatment. Unpaired two-tailed t-test. p-value = $1.9471 \times 10^{-13}$. n = 50, 138 cells for DMSO and BFA group respectively in (**a**); n = 5562, 6063, 5920, 4654 cells for DMSO, 1 μg/ml, 5 μg/ml and 10 μg/ml BFA group respectively for (**b** and **c**). DMSO: p-value = (vs. 1 μg/ml BFA) $1.6306 \times 10^{-35}$, (vs. 5 μg/ml BFA) $2.9606 \times 10^{-31}$ and (vs. 10 μg/ml BFA) $6.6553 \times 10^{-26}$ in (**b**); DMSO: p-value = (vs. 1/5/10 μg/ml BFA) 0 in (**c**). **d-f**, same as **a-c**, but in cells after 16 h DMSO or Tunicamycin (1 μg/ml, 5 μg/ml and 10 μg/ml) treatment. n = 35, 29 and 42 cells for DMSO, 5 μg/ml and 10 μg/ml Tunicamycin group respectively in (**d**). DMSO: p-value = (vs. 5 μg/ml Tunicamycin) $6.6064 \times 10^{-07}$ and (vs. 10 μg/ml Tunicamycin) $4.8959 \times 10^{-08}$. n = 5562, 6459, 5112 and 4761 cells for DMSO, 1 μg/ml, 5 μg/ml, 10 μg/ml and Tunicamycin group

in (**e** and **f**). DMSO: p-value = (vs. 1 μg/ml Tunicamycin) $1.6302 \times 10^{-10}$, (vs. 5 μg/ml Tunicamycin) $1.1949 \times 10^{-7}$ and (vs. 10 μg/ml Tunicamycin) $6.5454 \times 10^{-11}$ in (**e**); DMSO: p-value = (vs. 1/5/10 μg/ml Tunicamycin) 0 in (**f**). **g-i**, same as **a-c**, but in cells after 8 h or 12 h DMSO or Oligomycin (1 μM, 5 μM and 10 μM) treatment. n = 24, 42 and 27 cells for DMSO, 8 h and 12 h 10 μM Oligomycin group respectively in (**g**). DMSO: p-value = (vs. 10 μM Oligomycin 8 h) 0.9460 and (vs. 10 μM Oligomycin 12 h) 1. n = 5562, 8720, 6937 and 6680 cells for DMSO, 1 μM, 5 μM and 10 μM Oligomycin group in (**h** and **i**). DMSO: p-value = (vs. 1 μM Oligomycin) $2.0179 \times 10^{-5}$, (vs. 5 μM Oligomycin) $5.5789 \times 10^{-4}$ and (vs. 10 μM Oligomycin) $4.6361 \times 10^{-8}$ in (**h**); DMSO: p-value = (vs. 1/5/10 μM Oligomycin) 0 in (**i**). DMSO treated cells in (**b**, **c**, **e**, **f**, **h**, **i**) are from the same batch. One-way ANOVA and Scheffe's post hoc comparison in (**b** – **i**). Mean (dark) ± s.e.m. (shaded) in (**a**, **d**, **g**) and Mean (dark) ± s.d. (shaded) in (**b**, **c**, **e**, **f**, **h**, **i**).

# Reporting Summary

## Statistics

For all statistical analyses, confirm that the following items are present in the figure legend, table legend, main text, or Methods section.

| n/a | Confirmed | |
|---|---|---|
| ☐ | ☒ | The exact sample size (*n*) for each experimental group/condition, given as a discrete number and unit of measurement |
| ☐ | ☒ | A statement on whether measurements were taken from distinct samples or whether the same sample was measured repeatedly |
| ☐ | ☒ | The statistical test(s) used AND whether they are one- or two-sided *Only common tests should be described solely by name; describe more complex techniques in the Methods section.* |
| ☐ | ☒ | A description of all covariates tested |
| ☒ | ☐ | A description of any assumptions or corrections, such as tests of normality and adjustment for multiple comparisons |
| ☐ | ☒ | A full description of the statistical parameters including central tendency (e.g. means) or other basic estimates (e.g. regression coefficient) AND variation (e.g. standard deviation) or associated estimates of uncertainty (e.g. confidence intervals) |
| ☐ | ☒ | For null hypothesis testing, the test statistic (e.g. *F*, *t*, *r*) with confidence intervals, effect sizes, degrees of freedom and *P* value noted *Give P values as exact values whenever suitable.* |
| ☒ | ☐ | For Bayesian analysis, information on the choice of priors and Markov chain Monte Carlo settings |
| ☒ | ☐ | For hierarchical and complex designs, identification of the appropriate level for tests and full reporting of outcomes |
| ☐ | ☒ | Estimates of effect sizes (e.g. Cohen's *d*, Pearson's *r*), indicating how they were calculated |

*Our web collection on statistics for biologists contains articles on many of the points above.*

## Software and code

Policy information about availability of computer code

| | |
|---|---|
| Data collection | Data shown in Fig. 2g, 2h, 3e, 3f, Extended Data Fig. 3c, f-i and Extended Data Fig.7 d,e were captured by the NIS-Elements AR 5.21.03 controlled by a Ti2-E inverted microscope (Nikon) equipped with an LED light source (Lumencor Spectra X) and Hamamatsu ORCA-Flash4.0 V3 sCMOS camera. Data shown in Extended Data Fig.2 a,b were captured on a TIRF microscope (Nikon) equipped with 488/561/647 nm lasers and high numerical aperture objective in the Rockefeller University's Bio-Imaging Resource Center. Other imaging experiments were performed on a SoRa spinning-disk confocal microscope (Marianas system, 3i), equipped with a Zeiss Axio Observer 7 stand, ORCA-Fusion BT sCMOS Camera (Hamamatsu), CSU-W1 SoRa confocal scanner unit (Yokogawa), 405/445/488/514/561/637 nm LaserStack (3i) and an extra 'Vector' device for photomanipulation operated by the Slidebook 6 software. Cryo-patterned grids were visualized with a Titan Krios electron microscope (Thermo Fisher scientific) equipped with a field emission gun, a GIF Quantum LS postcolumn energy filter (Gatan), and a K3 summit electron detector (Gatan). The electron microscope was operated at 300 kv in nanoprobe mode at a magnification of 19,500x (pixel size of 4.53 Å at the specimen level). |
| Data analysis | Automated analysis of time-lapse imaging was performed using a custom MATLAB R2021 pipeline, reported in Bisaria, A., et al., Science 368, 1205-1210, (2020) and Chung et al., Mol Cell 76, 562-573,(2019) by our group. For significance test and non-linear regression analysis, data were processed by GraphPad Prism 9 or MATLAB R2021. The raw Cryo-ET tilt movies were motion and CTF-corrected in Warp 1.09, aligned with the AreTomo software package and further reconstructed with either weighted-back projection (WBP) in AreTomo or with simultaneous iterative reconstructive technique (SIRT) using Tomo3D. |

For manuscripts utilizing custom algorithms or software that are central to the research but not yet described in published literature, software must be made available to editors and reviewers. We strongly encourage code deposition in a community repository (e.g. GitHub). See the Nature Portfolio guidelines for submitting code & software for further information.

# Data

Policy information about availability of data

All manuscripts must include a data availability statement. This statement should provide the following information, where applicable:
- Accession codes, unique identifiers, or web links for publicly available datasets
- A description of any restrictions on data availability
- For clinical datasets or third party data, please ensure that the statement adheres to our policy

> The authors declare that all data supporting the findings of this study and source data in the excel file are available within the article and its supporting information files (extended data and supplementary videos).

# Research involving human participants, their data, or biological material

Policy information about studies with human participants or human data. See also policy information about sex, gender (identity/presentation), and sexual orientation and race, ethnicity and racism.

| | |
|---|---|
| Reporting on sex and gender | n/a |
| Reporting on race, ethnicity, or other socially relevant groupings | n/a |
| Population characteristics | n/a |
| Recruitment | n/a |
| Ethics oversight | n/a |

Note that full information on the approval of the study protocol must also be provided in the manuscript.

# Field-specific reporting

Please select the one below that is the best fit for your research. If you are not sure, read the appropriate sections before making your selection.

☒ Life sciences          ☐ Behavioural & social sciences          ☐ Ecological, evolutionary & environmental sciences

For a reference copy of the document with all sections, see nature.com/documents/nr-reporting-summary-flat.pdf

# Life sciences study design

All studies must disclose on these points even when the disclosure is negative.

| | |
|---|---|
| Sample size | Sample size for each experiment is indicated in figure legend. No statistical method was used to predetermine sample size and sample sizes were chosen based on pilot experiments and literature reports, sufficient for statistical analysis in our cell biology study. |
| Data exclusions | No data were excluded from our analysis. |
| Replication | All experiments were performed at least 3 times, but some data shown in the figures are one representative data from a single assay as indicated in the legend.<br>The data presented were successful in all 3 attempts or in the case of one failure, it was performed a fourth time to confirm the results to support conclusions stated in the manuscript. |
| Randomization | Randomization was not relevant for this study; Each experiment compared data sets from control conditions versus a perturbation(KD, inhibitor or over-experssion). Cells were selected based on their treatments. |
| Blinding | Investigators were not blinded for data collection, since the investigators were aware of sample and treatment allocations while setting up the experiments. |

# Reporting for specific materials, systems and methods

We require information from authors about some types of materials, experimental systems and methods used in many studies. Here, indicate whether each material, system or method listed is relevant to your study. If you are not sure if a list item applies to your research, read the appropriate section before selecting a response.

## Materials & experimental systems

| n/a | Involved in the study |
|-----|----------------------|
| ☐ | ☒ Antibodies |
| ☐ | ☒ Eukaryotic cell lines |
| ☒ | ☐ Palaeontology and archaeology |
| ☒ | ☐ Animals and other organisms |
| ☒ | ☐ Clinical data |
| ☒ | ☐ Dual use research of concern |
| ☒ | ☐ Plants |

## Methods

| n/a | Involved in the study |
|-----|----------------------|
| ☒ | ☐ ChIP-seq |
| ☒ | ☐ Flow cytometry |
| ☒ | ☐ MRI-based neuroimaging |

## Antibodies

| | |
|---|---|
| Antibodies used | Phospho-Tyrosine (P-Tyr-1000) MultiMab Rabbit mAb Mix (1:500, Cell Signaling Technology, #8954) and CLIMP63 monoclonal antibody (G1/296) (1:500, Enzo Life Sciences, ENZ-ABS669-0100). Anti-ESYT1 antibody (1:200, Sigma, HPA076926), Anti-ESYT2 antibody (1:1000, Sigma, HPA002132), PTP1B antibody (1:1000, BD Bioscience, 610139), RTN4 antibody (1:1000, Thermo Scientific, #MA5-32763). Secondary antibodies included goat anti-rabbit IgG(H+L) Alexa Fluor 568, Invitrogen (1:2000, Thermo Scientific, A-11011), goat anti-rabbit IgG(H+L) Alexa Fluor 647, Invitrogen (1:2000, Thermo Scientific, A-21245) and goat anti-rabbit IgG(H+L) Alexa Fluor 700, Invitrogen (1:2000, Thermo Scientific, A-21038). |
| Validation | All antibodies are commercially available and validated by the authors using western blot or immunofluorescence assay shown in the paper. All antibodies were used in multiple experiments to detect the indicated target protein giving results consistent to the expected moelcular weight, subcellular localization or signal changes after inhibitor treatment or siRNA knockdown. |

## Eukaryotic cell lines

Policy information about cell lines and Sex and Gender in Research

| | |
|---|---|
| Cell line source(s) | hTERT RPE-1 cells (here referred to as RPE-1) (ATCC CRL-4000), HUVEC/TERT2 cells (ATCC CRL-4053), BJ-5ta cells (ATCC CRL-4001) and 293T cells (Takara Bio 632180) were used in this study. |
| Authentication | All cell lines were received directly from the supplier and other stable cell lines were constructed based on the original cell lines. |
| Mycoplasma contamination | All cell lines were verified to be mycoplasma free. |
| Commonly misidentified lines (See ICLAC register) | No commonly misidentified cell lines were used. |

