## [Peer Review File · Nature]

Manuscript Title: Endoplasmic Reticulum-Plasma Membrane contact gradients direct cell migration.

Editorial Notes:

Redactions – unpublished data

Reviewer Comments & Author Rebuttals

Reviewer Reports on the Initial Version:

Referees' comments:

Referee #1 (Remarks to the Author):

The study by Gong et al. reports that membrane contact sites between the ER and the PM are more stable at the back of migrating cells due to a differential recruitment of ER-shaping proteins. The persistent contacts restrict the activity of an ER-bound phosphatase at the back of cells, acting as inhibitory signalling platforms that preserve the memory of the directionality of migrating cells. The concept is elegant but inferred from live-cell imaging data with reporter probes that have biological activity. More solid ultrastructural and mechanistic evidence is required to substantiate the claim that a different shape and dynamics of cortical ER structures control the direction of cell migration.

1. The study relies extensively on live-imaging of cells ectopically expressing an ER-bound protein that binds phosphoinositides with high avidity. This “membrane-attached peripheral ER” (MAPPER) is not simply labelling endogenous contact sites as assumed, but is an ER-PM tether that can generate new contact sites and modify the morphology of native cortical ER structures. Several reports indicate that MAPPER proteins induce the formation of abnormally large contact sites (PMID: 29138405; PMID: 29469807; PMID: 35191477). Using MAPPER as unique tool to track the location and dynamics of endogenous MCS is insufficient, because 1) MAPPER expression might stabilize and extend native cortical ER structures, therefore artificially generating the persistent contacts reported here, and 2) the membrane attachment of the ER-bound probe might report the different lipid composition between the front and back of migrating cells, and not the presence of native cortical structures at specific locations.

2. Individual contact sites are quantified from the patterns of diffraction-limited fluorescence signals and their shapes inferred from the differential recruitment of ER-shaping proteins. The individual structures cannot be resolved by light microscopy, and the recruitment of CLIMP63 and RTN4 to different cellular locations do not prove that native structures exhibit the morphology enforced by these ER-shaping proteins prior to their ectopic expression. This question can only be solved by electron microscopy. Ultrastructural evidence is required to establish that endogenous ER-PM contact sites form preferentially at the back of migrating cells and that their morphology differ from the contacts forming at the migrating front.

3. The study is essentially descriptive and lacks mechanistic intervention, besides the silencing of extended synaptotagmins. Optogenetic tools exist that can enforce contact sites formation at specific locations inside cells (PMID: 26322679; PMID: 28959426; PMID: 33174843; PMID: 32098964). ER-PM contacts generated artificially at the front or disrupted at the back of cells would

be predicted to reverse the direction of cell migration, providing direct evidence for the mechanism proposed.

4. The evidence that contact sites have a different molecular composition at the front and back is insufficient. Only endogenous CLIMP63 immunoreactivity is shown in Fig. S5 and the antibody does not decorate contact sites but labels perinuclear structures on both sides of the nucleus. Additional evidence is required to show that different endogenous ER-shaping proteins are enriched at the two opposite poles of migrating cells.

5. Fig. 2C: The images showing the redistribution of contact sites during collective migration of RPE1 cells are not very convincing, with only one cell outlined. Could this be tested in a wound healing assay?

Referee #2 (Remarks to the Author):

The manuscript describes a novel role for ER-PM contacts in the control of cell migration. The control mechanism is based on a polarized distribution of ER types, tubular ER with RTKs that increase PIP3 at the leading edge and sheet-like ER with PTP1B phosphatase at the trailing edge. In addition, the authors found small, labile ER-PM contacts at the leading edge and large, stable ones at the back. The gradient was established through RTN4 at the front and CLIMP63 at the back. The authors used micropatterned linear stripes of fibronectin to discover this mechanism. On this system, the cells start to migrate randomly in one direction upon the presence of serum or EGF. The front was characterized by high pTyr labeling and low levels of ER-PM contacts. The back showed large ER-PM contact sites. Movement could be disrupted in the presence of LY294002. The same movement characteristics were seen in different cell lines, in populations of multiple cells and in 3D. Knockdown of E-Syts reduced the speed of movement. The same was seen upon inhibition of PTP1B. Importantly, the ER-PM contacts at the trailing edge remained stable relative to the ECM, even if the direction of movement was reversed.

This is a beautiful study that requires little modification to be convincing. In particular, the study would benefit from putting it into context with ER functions such as origin of protein trafficking, protein folding and calcium storage, to name but a few. Another question concerns the need for energy. Answering this latter question would significantly increase the significance of the findings for cancer biology. Both questions can be addressed with easy assays. Despite these illustrative additions, this is an important study that has clear-cut, easy conclusions and provides significant new knowledge in cell movement.

1. The data using the PTP1B siRNA in extended Figure 3a/b is not convincing. The graph suggests a much smaller effect than what the figure shows.
2. The extent of knockdown achieved in the experiments remains unclear. Normally, this should be assayed by Western blot. In the experimental setup, the authors produce 2-3000 cells, which should allow for the production of one control Western blot well. Alternatively, IF could be used to give an idea how efficient the knockdown was.
3. It would be interesting to see how ER homeostasis affects this mechanism. This could be easily assayed via Brefeldin A, tunicamycin or cyclopiazonic acid, to give an idea how membrane

trafficking, unfolded proteins or calcium influence this phenomenon.

4. Is mitochondrial or glycolytic energy necessary for the establishment or maintenance of the ER-PM patches? Admittedly, blocking both energy sources may stop movement altogether, but potentially the authors could switch to galactose medium or hypoxia/oligomycin to increase significance of this mechanism in cancer, as stated by the authors in the introduction.

Referee #3 (Remarks to the Author):

Gong et al. in the Meyer lab reports the remarkable discovery that a gradient in close contacts between the ER and PM is important to establish the polarity of migrating cells. While close contacts have been known to be important in store operated calcium entry, for example in muscle contraction, a role in cell migration is unexpected and exciting. The system of cell migration confined to narrow cell-diameter scale stripes is clever and allows migration polarity to be accurately measured on individual cells and averaged over significant numbers of cells. MAPPER from the Liou lab is used to measure close contacts and is compared to generalized markers of the ER and PM in the same cell. The authors should do a better job of explaining MAPPER. I had to read the Liou paper to understand that MAPPER is concentrated at close contacts because it interacts with both the PM and the ER; at first I mistakenly thought it was a distance reporter using FRET. Remarkably the ER-PM contact gradient is stable in cells, it persists for over 20 min during inhibition of PI3K, and cells take even longer to reverse their polarity when they reach the ends of stripes. This polarity/motility direction change almost perfectly correlates with change in close contact gradient. An unanswered question is how contact gradient correlates with MTOC orientation during these pauses and direction changes. However, the authors do suggest a relation to microtubule guidance of ER. The study is carefully done and there are a number of important corollary discoveries. Using inhibitors and siRNA, the authors demonstrate that the ER resident phosphatase PTP1B acts at the back of cells to lower tyrosine phosphates there. Remarkably, the authors also used split GFP to demonstrate a direct interaction between EGFR and PTP1B at the back of migrating cells. The authors also showed that a marker of ER curvature localizes to the front of migrating cells and a marker of flat, sheet-like ER localizes to the back of migrating cells.

They also show that the steepness of the gradient correlates with speed of migration, within the population of cells in a given cell type. From the text, though, it sounds like they are making a more sweeping generalization about speed among different cell types. What about fish scale keratinocytes and leukocytes that migrate fast? Do they have steeper gradients than fibroblasts? As the amount of work is already extensive, they do not need to test other cell types, but they should qualify the claim.

This work is orthogonal to an already extensive literature on Rho family GTPases and PIP lipids in cell polarity and cell migration, and adds a significant new dimension to our understanding of the regulation of cell migration. However, as they point out, the close contact gradient is much more stable than these other regulators in governing migration. And could they say something about the MTOC? Is it also well correlated with migration direction and does it also switch orientation at the same time as the close contact gradient? Is MTOC orientation better correlated than that of the nucleus?

As the ER is an important site for phosphatidylinositol lipid synthesis, this work might lead to further

discoveries in the future about the roles of PIP2 and PIP3 in regulating cell polarization, which seems important but is still rather murky. Perhaps they could add a few more thoughts on this to their sentences on lines 317-319.

Tim Springer

Referee #4 (Remarks to the Author):

This study reports a novel mechanism controlling the cell protrusion at the front and suppression of signalling at the back of cell migrating in single or collective migration. Using a live-cell ER-PM reporter, it is shown that it is regulated by PM-ER contact gradients, small and unstable at the front and large, stable ones at the back. It is proposed that ER-PM contact gradients between front and back of the cells shape the ER morphology (sheets/tubules) through the activity/expression of the two molecules RTN4 and CLIMP63.

How does the ER-PM gradient connect with the actin cytoskeleton /associated molecules to lead to cell protrusion at the front? The authors mention connection with microtubules in the discussion. Could this hypothesis be tested?

The techniques employed appear mostly appropriate and a number of results are exciting. However, there are a number of issues listed below:

Results displayed Figure 1 are clear but there is no statistics.

Figure 2 panels a-e are too descriptive with only one cell shown with a certain phenotype. These panels need to include quantitative and statistics data. Panel C is not convincing. It is not clear these cells are ongoing collective migration and the gradient of ER-PM is not really visible. Why would it show in only one cell? Panel f and g are quantified but no statistics is provided. Are cells migrating in the amoeboid mode in Panel d? If so, this finding should be expanded as this could mean this PM-ER contact gradient plays a role in mesenchymal and amoeboid migration.

Figure 4. Quantitative results with statistics should be provided for Fig 4 a-c. The kymographs in d and e present the results of how many cells?

Figure 5. Although the results presented panels b and c are clearly visible, 5d should provide statistics from 3 independent experiments with at least 30 cells per condition and per experiments analysed.

The group of Jennifer Lippincott-Schwartz has shown that ER organisation in sheet in tubules is an "old fashion view" of ER morphology and in fact the so-called sheets are packed tubules (Nixon-Abell et al, Science 2016). This manuscript should integrate these findings. Provide some references lines 236 and 237 regarding RTN4 and CLIMP63 roles in the formation of ER tubules and sheets. Is there an explanation why the effect of siCLIMP63 is much stronger than the one of siRTN4? Although the data presented panels e-i are quite convincing, they only suggest the change in ER morphology. The data should be backed up with appropriate imaging of ER.

Additional comments

Indicate in the result part what is the “expressed PM marker” (line 72).

Staining for phospho-tyrosine is not specific of RTK and thus does not determine “the spatial organisation of RTK signalling” per se (line 78).

Extended data Legend of Figure 1: Details of the “CaaX signal” in RPE-1 cells is missing. Clear specification of the cells must be provided (“expressing a mTurquoise-CaaX PM marker” as in Figure 1 legend).

Fig 4a. There is a typo in the title “stionary.

Author Rebuttals to Initial Comments:

Referees' comments:

Referee #1 (Remarks to the Author):

The study by Gong et al. reports that membrane contact sites between the ER and the PM are more stable at the back of migrating cells due to a differential recruitment of ER-shaping proteins. The persistent contacts restrict the activity of an ER-bound phosphatase at the back of cells, acting as inhibitory signalling platforms that preserve the memory of the directionality of migrating cells. The concept is elegant but inferred from live-cell imaging data with reporter probes that have biological activity. More solid ultrastructural and mechanistic evidence is required to substantiate the claim that a different shape and dynamics of cortical ER structures control the direction of cell migration.

1. The study relies extensively on live-imaging of cells ectopically expressing an ER-bound protein that binds phosphoinositides with high avidity. This “membrane-attached peripheral ER” (MAPPER) is not simply labelling endogenous contact sites as assumed but is an ER-PM tether that can generate new contact sites and modify the morphology of native cortical ER structures. Several reports indicate that MAPPER proteins induce the formation of abnormally large contact sites (PMID: 29138405; PMID: 29469807; PMID: 35191477). Using MAPPER as unique tool to track the location and dynamics of endogenous MCS is insufficient, because 1) MAPPER expression might stabilize and extend native cortical ER structures, therefore artificially generating the persistent contacts reported here, and 2) the membrane attachment of the ER-bound probe might report the different lipid composition between the front and back of migrating cells, and not the presence of native cortical structures at specific locations.

Response: Thank you for bringing up these valid points. We should have discussed the reports mentioned by the reviewer and the potential contributions from lipid gradients. We are separately discussing in 3 sections the 1) MAPPER expression, 2) lipid composition and 3) new experiments.

1) We noticed as well that the number of ER-PM contacts increases by overexpression of high levels of the MAPPER sensor (we similarly observed earlier that high expression of STIM1 beyond its normal level can promote ER-PM contact formation). Such effects from overexpression can be reduced or eliminated by using sensitive cameras and imaging conditions, which allows for the sorting of reporter cells that have low MAPPER (or STIM1) expression. We would like to note that we can see the front-to-back gradient in MAPPER even in cells that have minimal MAPPER expression. To specifically address the three papers: In PMID 29138405, they showed that overexpression of MAPPER with an mCherry tag results in a higher expansion of the contacts, compared to GFP-MAPPER. We think this may in part be explained by a requirement of a higher expression level of mCherry tag for an equivalent signal/noise compared to the GFP version (so we used in our study the YFP version). In PMID 29469807, they described the overexpression issue in the point-to-point response letter without showing the data, so we don't have detailed information to comment on. In PMID 35191477, they did a thorough comparison of ER-PM distance and length between the control group and MAPPER overexpression group at the rest state versus the ER calcium depletion state. As they stated, at the rest state, MAPPER-S has no effect on cortical ER, while MAPPER-L induced the formation of cER cisternae longer than 300 nm in some cells. However, there

is no significant difference in ER-PM contact distance and length between control cells and MAPPER-S or MAPPER-L overexpressing cells (see below, Fig. R 1a).

We would like to add that in our study, the MAPPER signals are always compared from the front to back of the same cell. If there would be an increase in the contact number caused by the MAPPER expression, it would be expected to modify all contacts, rather than being the cause of the gradient we are observing in the migrating cells.

Based on these considerations, we think that MAPPER can be used as an ER-PM contact marker but only when cells are sorted for low expression levels. We are now including this important issue in the text.

2) The reviewer is making an interesting point about the known difference in the lipid composition in the front and back. We do think that the lipid difference is likely making a contribution to the contact site gradient, given that several highly conserved ER-PM tethers are targeted to the PM, particularly via interaction with PI(4,5)P2 (eg, the extended synaptotagmins (E-Syt)).

[REDACTED]

3) To directly test for the ER-PM contact gradient without expression of MAPPER, we have performed additional experiments which are supporting our observation of ER-PM contacts gradient in migrating cells. a) We compared to ER-PM contact area of the front versus back ER in Cryo-ET images of wild type cells (without MAPPER): we found there were about 12 times more back ER close to the PM within a 40 nm gap distance than the front ER (Fig. R1 b-d, Fig. 1 j-l); b) In addition, we used TIRF image analysis of endogenous E-Syt1 staining in wild type cells, which also revealed a back-to-front ER-PM contact gradient (Fig. R1 e, f, Extended Fig. 2 a, b).

Fig. R1. EM-PM contacts gradient in migrating cells by Cryo-ET and endogenous E-Syt1 staining. Shown in Fig. 1 j-l and Extended Fig. 2 a, b. **a**, Significance difference test in ER-PM contact distance and length between control cells and MAPPER-S or MAPPER-L overexpressing cells. Modified from PMID:35191477. **b**, Segmentation and 3D rendering of ER and PM from the front of a migrating cell. MCS indicates the site where the distance between the ER and PM is within 40 nm. **c**, Same as **(b)** for the back of a migrating cell. **d**, Quantification of the ER-PM contact area of the front versus back ER in Cryo-ET images. Any ER apposing to the PM within a 40 nm gap distance is counted. 7 front and 7 back tomograms were used for quantification. Paired t-test, **** $p < 0.0001$. **e**, Representative TIRF image of fixed RPE-1 cells stably expressing iRFP-CaaX (yellow) and EGFP-Sec61β (magenta) after E-Syt1 staining (cyan). Right, E-Syt1/CaaX and E-Syt1/Sec61β ratio color images. Scale of E-Syt1/CaaX and E-Syt1/Sec61β is normalized for maximal signal range as shown in corresponding color bars. **f**, Mean E-Syt1/CaaX and E-Syt1/Sec61β gradient. Bold line and shaded area are mean \pm s.e.m., $n=20$ cells.

2. Individual contact sites are quantified from the patterns of diffraction-limited fluorescence signals and their shapes inferred from the differential recruitment of ER-shaping proteins. The individual structures cannot be resolved by light microscopy, and the recruitment of CLIMP63 and RTN4 to different cellular locations do not prove that native structures exhibit the morphology enforced by these ER-shaping proteins prior to their ectopic expression. This question can only be solved by electron microscopy.

Ultrastructural evidence is required to establish that endogenous ER-PM contact sites form preferentially at the back of migrating cells and that their morphology differ from the contacts forming at the migrating front.

Response: We agree that additional structural data is helpful. We have therefore conducted Cryo-ET experiments in the front and back of wild type cells to investigate their native ER structures. After reconstruction and segmentation, we clearly observe more ER tubules in the front, in contrast to the ER sheet in the back (Fig. R 2a, b, Fig. 5e, f). The surface morphometric analysis (measuring a curvature parameter) further revealed that the ER in the front is about 2.23-fold more curved than in the back (Fig. R 2c, Fig. 5g), consistent with the visually observed differences between the front and back ER. We further quantified the relative area of the ER that is parallel to the PM within a 40 nm gap distance. This analysis revealed that there is about 12 times more back ER forming ER-PM contacts than in the front ER, consistent with the ER-PM contacts gradient data observed with the MAPPER sensor. By comparing the curvature parameter of the ER specifically at the MCSs, we found that the front ER at MCSs is almost 10-fold more curved than the back ER at MCSs (Fig. R 2d), further suggesting the importance of ER curvature in the MCS formation.

Fig. R2. Cryo-ET analysis of the front and back of migrating cells. Shown in Fig. 5e-g. a, Segmentation and 3D rendering of ER and PM from the front of a migrating cell. b, Same as (a) for the back of a migrating cell. MCS indicates the site where the distance between the ER and PM is within 40 nm. c, Surface morphometric analysis of the front and back ER in Cryo-ET images to quantify their curvedness. 7 front and 7 back tomograms were used for quantification. Unpaired two-tailed Student's t-test, *p = 0.0006. d, Same as (c) for the ER at ER-PM contact sites.**

3. The study is essentially descriptive and lacks mechanistic intervention, besides the silencing of extended synaptotagmins. Optogenetic tools exist that can enforce contact sites formation at specific locations inside cells (PMID: 26322679; PMID: 28959426; PMID: 33174843; PMID: 32098964). ER-PM contacts generated artificially at the front or disrupted at the back of cells would be predicted to reverse the direction of cell migration, providing direct evidence for the mechanism proposed.

Response: Thank you for your suggestion of using an optogenetic strategy to perturb the ER-PM gradient. We followed the reviewer's recommendation and overexpressed CRY2-mCherry-Sec61 β and CaaX-iRFP-CIBN in the RPE1 cells to test whether increased ER-PM contact formation in the front is changing the direction of cell migration (Fig. R3a, Fig. 2 a). Before light activation to dimerize the ER and PM localized constructs, which triggers the formation of new ER-PM contacts, CRY2-mCherry-Sec61 β was evenly distributed at the ER as expected (from -10 min to 0 min) and the cell was moving towards the left. Upon blue light activation in the front region, CRY2-mCherry-Sec61 β positive puncta gradually increased, indicating that ER-PM contacts were growing. Markedly, along with the new puncta formation, membrane protrusions in the initial front were gradually retracted and meanwhile, small new protrusions were made from the original back, which ultimately resulted in a reversal of cell migration, with the cell moving toward the right (Fig. R3b, Fig. 2 b). Among all the 39 illuminated cells from 3 independent experiments, about 50.6% of cells' direction could be reversed and 23% triggered instead an extended side protrusion between the front and back of the cell (Fig. R3c and d, Fig. 2 c and Extended Fig. 3 a). This data strongly suggests that the ER-PM contact gradient can regulate the direction of cell migration.

Fig. R3. Optogenetic control of cell direction. Shown in Fig. 2 a-c and Extended Fig. 3 a. **a**, Schematic of optogenetic tools to establish contacts between ER and PM. CRY2-mCherry-Sec61 β and CaaX-iRFP-CIBN are stably expressed in the RPE1 cells. Blue light illumination induces the CRY2-CIBN complex formation to trigger ER-PM contact formation. **b**, Cell movement before and during blue light illumination. White arrows indicate the moving direction and white boxes indicate the illuminated region. Blue light was on every 2s. **c**, Percentage of cells with reversed behavior before and after blue light illumination. Cells in the control group were stably expressing CaaX-iRFP and Sec61 β -EGFP. n=29 for control group and 39 for optogenetic group from 3 experiments. *p= 0.0113. **d**, Three different cell behaviors after blue illumination in control and optogenetic group cells.

4. *The evidence that contact sites have a different molecular composition at the front and back is insufficient. Only endogenous CLIMP63 immunoreactivity is shown in Fig. S5 and the antibody does not decorate contact sites but labels perinuclear structures on both sides of the nucleus. Additional evidence is required to show that different endogenous ER-shaping proteins are enriched at the two opposite poles of migrating cells.*

Response: Thank you for bringing up another relevant point about the composition of the contacts in the front and back. To address this point, we masked out the membrane contact sites (MCSs) from the cell and assessed the distribution of CLIMP63 and RTN4 in and out of the MCSs. It turns out, as the reviewer pointed out, the endogenous CLIMP63 signal doesn't specifically decorate contact sites, but shows a similar enrichment at the back both in and out of the MCSs, indicating a general enrichment of CLIMP63 in the sheet-like ER around the nucleus at the back (Fig. R 4a [REDACTED]). This is also true for RTN4, which shows a general enrichment in the tubular ER at the front both in and out of the MCSs (Fig. R 4b[REDACTED]). This piece of data further supports that there are two distinct ER morphologies at the front and back, which is also consistent with the Cryo-ET data. We are cognizant that RTN4 and CLIMP63, along with other ER shaping proteins, are important for building and maintaining the tubular and sheet ER but we are using the localization of RTN4 and CLIMP63 here mostly as markers for the different ER morphologies in the front and back. We do not think (and our data do not support) that these ER-shaping proteins are specifically enriched at front and back MCSs to help organize the MCS formation per se.

[REDACTED]

[REDACTED]

5. Fig. 2C: The images showing the redistribution of contact sites during collective migration of RPE1 cells are not very convincing, with only one cell outlined. Could this be tested in a wound healing assay?

Response: Thank you for this comment. We performed a wound healing scratch assay to test for ER-PM contacts in collectively migrating cells. A series of still images (0h, 2h, 4h and 6h after wound) from a time-course video of the ER-PM contact gradients in the confluent cells are shown to prove their ongoing collective migration. Meanwhile, a representative leader cell and follower cell are separately zoomed in to reveal the establishment/distribution of ER-PM contact gradients across the time. The video is also provided in Supplementary Video 2. At the initial phase (0 h) when a scratch was made from a sheet of cells under confluent state, cells had only few contact sites and no obvious ER-PM contact gradients could be observed in the immobile cells. When cells were moving towards the wound, more and more contact sites were gradually formed in the cells and a clear back-to-front gradient developed in both the leader and follower cells (Fig. R 5 and Fig. 1m, Extended Fig. 2 m,n and Supplementary Video 2).

Fig. R5. Distribution of ER-PM contacts in collectively migrating RPE1 cells during wound closure at 0h, 2h, 4h and 6h. Shown in Fig. 1m, Extended Fig. 2 m,n and Supplementary Video 2. a, Time-course of the collectively migrating RPE-1 cells stably expressing mTurquoise-CaaX and MAPPER-mVenus in the wound closure assay. A representative leader cell and follower cell are separately zoomed in to reveal the distribution of ER-PM contact gradients across the time. PM are indicated by the white dash boundary. **b,c,** Mean MAPPER/CaaX gradients in leader cells (**b**, $n = 46$ cells) and follower cells (**c**, $n = 33$ cells) during collective cell migration. Mean (dark green) \pm s.e.m. (shaded green).

Referee #2 (Remarks to the Author):

The manuscript describes a novel role for ER-PM contacts in the control of cell migration. The control mechanism is based on a polarized distribution of ER types, tubular ER with RTKs that increase PIP3 at the leading edge and sheet-like ER with PTP1B phosphatase at the trailing edge. In addition, the authors found small, labile ER-PM contacts at the leading edge and large, stable ones at the back. The gradient was established through RTN4 at the front and CLIMP63 at the back. The authors used micropatterned linear stripes of fibronectin to discover this mechanism. On this system, the cells start to migrate randomly in one direction upon the presence of serum or EGF. The front was characterized by high pTyr labeling and low levels of ER-PM contacts. The back showed large ER-PM contact sites. Movement could be disrupted in the presence of LY294002. The same movement characteristics were seen in different cell lines, in populations of multiple cells and in 3D. Knockdown of E-Syts reduced the speed of movement. The same was seen upon inhibition of PTP1B. Importantly, the ER-PM contacts at the trailing edge remained stable relative to the ECM, even if the direction of movement was reversed. This is a beautiful study that requires little modification to be convincing. In particular, the study would benefit from putting it into context with ER functions such as origin of protein trafficking, protein folding and calcium storage, to name but a few. Another question concerns the need for energy. Answering this latter question would significantly increase

the significance of the findings for cancer biology. Both questions can be addressed with easy assays. Despite these illustrative additions, this is an important study that has clear-cut, easy conclusions and provides significant new knowledge in cell movement.

Response: We are grateful for the reviewer's supportive and constructive comments. The requested changes were very helpful.

1. The data using the PTP1B siRNA in extended Figure 3a/b is not convincing. The graph suggests a much smaller effect than what the figure shows.

Response: We apologize for the lack of clarity. The statistical data in the original Extended Fig.3b (now Extended Fig.4c) indicates that the ER-PM contact gradient is not lost after knockdown of PTP1B. This is also consistent with the MAPPER/CaaX ratio data after the PTP1B inhibitor treatment, shown in Fig.3 a-c. When we quantified the MAPPER/CaaX ratio across the cell, we normalized the MAPPER intensity to its median level in each cell to correct for variation in sensor expression among cells. Thus, cells can have the same normalized MAPPER/CaaX gradient but can have overall more or less averaged MAPPER intensity. To clarify this point, we have now correctly labelled the y axis in the extended Fig.3b (now Extended Fig.4c) as the normalized MAPPER/CaaX ratio.

2. The extent of knockdown achieved in the experiments remains unclear. Normally, this should be assayed by Western blot. In the experimental setup, the authors produce 2-3000 cells, which should allow for the production of one control Western blot well. Alternatively, IF could be used to give an idea how efficient the knockdown was.

Response: We have now added the data on how efficient the different siRNAs are. We used western blot analysis to detect CLIMP63, RTN4, PTP1B and E-Syt2 protein level in the cell lysates from WT, control and target siRNA transfected cells (Fig. R 6a-d, Extended Data Fig. 3 d, 4 a, and 7 a, b). All the knockdown effects were significant. Single-cell immunofluorescence analysis also showed a significant knockdown of E-Syt1 protein and a slight compensation from E-Syt2 (Fig. R 6e, Extended Data Fig. 3 c).

Fig. R6. siRNA efficiency test. Shown in Extended Data Fig. 3 c, d, 4 a, and 7 a, b. **a**, Western-blot analysis of cell lysates from WT, control and target siRNA transfected cells. E-Syt2 and Actin were detected. **b-d**, Same as (a) for RTN4 (b) or PTP1B (c) or E-Syt2 (d) and Actin detection. **e**, Single-cell immunofluorescence analysis of wild type cells and cells with indicated siRNA transfection. Wild type and siRNA-transfected cells were fixed for E-Syt1 immunostaining. N = 16438 cells for wild type cells, n = 13094 for control siRNA group, n = 14758 cells for E-Syt1 siRNA group and n = 14153 cells for E-Syt2 siRNA group. One-way ANOVA and scheffe's post hoc comparison. ***p < 0.0001 (control vs. E-Syt1 siRNA).

3. It would be interesting to see how ER homeostasis affects this mechanism. This could be easily assayed via Brefeldin A, tunicamycin or cyclopiazonic acid, to give an idea how membrane trafficking, unfolded proteins or calcium influence this phenomenon.

Response: Thank you for the interesting suggestion. We tested the effect of Brefeldin A (BFA), tunicamycin (Tuni) or cyclopiazonic acid (CPA) on cell migration, measuring mean square displacement (MSD), cell speed as well as the ER-PM contact gradients. Both the Brefeldin A (BFA) and tunicamycin (Tuni) treatment significantly blocked cell migration and had reduced ER-PM contact gradients (Fig. R 7 a-f, Extended Data Fig. 7 j - o), suggesting a potential role of membrane trafficking and unfolded protein response in the formation of ER-PM contact gradients. In contrast, cells after cyclopiazonic acid (CPA) treatment shows no striking difference in cell migration and ER-PM contact gradients, compared to the DMSO treatment (Fig. R 7 g-i). We now added these results to the revised manuscript.

Fig. R7. Effects of the ER homeostasis related drugs on cell migration and ER-PM contact gradients. Shown now as Extended Data Fig. 7 j-o. **a-c**, Mean square displacement (**a**), mean cell speed (**b**) and mean gradients of MAPPER/CaaX ratio (**c**), analysis of RPE-1 after DMSO or BFA (10 $\mu\text{g/ml}$) treatment. $N = 5562$ cells, DMSO; $n = 6063$ cells, 1 $\mu\text{g/ml}$ BFA group; $n = 5920$ cells, 5 $\mu\text{g/ml}$ BFA group and $n = 4654$ cells, 10 $\mu\text{g/ml}$ BFA group for (**a** and **b**). $***p < 0.0001$. $N = 53$ cells, DMSO; $n = 142$ cells, BFA group for (**c**); **d-f**, same as **a-c**, but in cells after DMSO or Tunicamycin (5 $\mu\text{g/ml}$ and 10 $\mu\text{g/ml}$) treatment. $N = 5562$ cells, DMSO; $n = 6459$ cells, 1 $\mu\text{g/ml}$ Tunicamycin group; $n = 5112$ cells, 5 $\mu\text{g/ml}$ Tunicamycin group; $n = 4761$ cells, 10 $\mu\text{g/ml}$ Tunicamycin group for (**d** and **e**). $***p < 0.0001$. $N = 35$ cells, DMSO; $n = 35$ cells, 5 $\mu\text{g/ml}$ Tunicamycin group; $n = 47$ cells, 10 $\mu\text{g/ml}$ Tunicamycin group for (**f**). **g-i**, same as **a-c**, but in cells after DMSO or cyclopiazonic acid (5 μM , 10 μM and 30 μM) treatment.

4. Is mitochondrial or glycolytic energy necessary for the establishment or maintenance of the ER-PM patches? Admittedly, blocking both energy sources may stop movement altogether, but potentially the authors could switch to galactose medium or hypoxia/oligomycin to increase significance of this mechanism in cancer, as stated by the authors in the introduction.

Response: We now also tested the potential contributions from different energy sources. As suggested by the reviewer, oligomycin was used to block mitochondrial ATP synthesis. Consistent with a role of mitochondria, oligomycin treatment compromised RPE1 cell migration. However, there was no significant change in the ER-PM contact gradients, indicating that the effect on the cell migration by oligomycin might be related to actin-mediated membrane protrusion rather than the ER-PM contact gradients. Given the differences in cell metabolism between cancer cell lines and RPE1 cells, there are interesting follow-up questions to understand the potentially unique contributions from cancer cell metabolism to ER-PM contact and actin regulation.

Fig. R8. Effects of Oligomycin treatment on cell migration and ER-PM contact gradients. a-c, Mean square displacement (a), mean cell speed (b) and mean gradients of MAPPER/CaaX ratio (c), analysis of RPE-1 after DMSO or Oligomycin (1 μM , 5 μM and 10 μM) treatment.

Referee #3 (Remarks to the Author):

Gong et al. in the Meyer lab reports the remarkable discovery that a gradient in close contacts between the ER and PM is important to establish the polarity of migrating cells. While close contacts have been known to be important in store operated calcium entry, for example in muscle contraction, a role in cell migration is unexpected and exciting. The system of cell migration confined to narrow cell-diameter scale stripes is clever and allows migration polarity to be accurately measured on individual cells and averaged over significant numbers of cells. MAPPER from the Liou lab is used to measure close contacts and is compared to generalized markers of the ER and PM in the same cell. **(i) The authors should do a better job of explaining MAPPER.** I had to read the Liou paper to understand that MAPPER is concentrated at close contacts because it interacts with both the PM and the ER; at first I mistakenly thought it was a distance reporter using FRET. Remarkably the ER-PM contact gradient is stable in cells, it persists for over 20 min during inhibition of PI3K, and cells take even longer to reverse their polarity when they reach the ends of stripes. This polarity/motility direction change almost perfectly correlates with change in close contact gradient. **(ii) An unanswered question is how contact gradient correlates with MTOC orientation during these pauses and direction changes.** However, the authors do suggest a relation to microtubule guidance of ER. The study is carefully done and there are a number of important corollary discoveries. Using inhibitors and siRNA, the authors demonstrate that the ER resident phosphatase PTP1B acts at the back of cells to lower tyrosine phosphates there. Remarkably, the authors also used split GFP to demonstrate a direct interaction between EGFR and PTP1B at the back of migrating cells. The authors also showed that a marker of ER curvature localizes to the front of migrating cells and a

marker of flat, sheet-like ER localizes to the back of migrating cells. They also show that the steepness of the gradient correlates with speed of migration, within the population of cells in a given cell type. From the text, though, it sounds like they are making a more sweeping generalization about speed among different cell types. **(iii) What about fish scale keratinocytes and leukocytes that migrate fast? Do they have steeper gradients than fibroblasts? As the amount of work is already extensive, they do not need to test other cell types, but they should qualify the claim.** This work is orthogonal to an already extensive literature on Rho family GTPases and PIP lipids in cell polarity and cell migration, and adds a significant new dimension to our understanding of the regulation of cell migration. However, as they point out, the close contact gradient is much more stable than these other regulators in governing migration. **(iv) And could they say something about the MTOC? Is it also well correlated with migration direction and does it also switch orientation at the same time as the close contact gradient? Is MTOC orientation better correlated than that of the nucleus?**

Response: We thank the reviewer for appreciating this new mechanism of cell migration regulation. We believe that the specific comments helped us to improve the strength and clarity of the manuscript.

(i) We are now introducing the MAPPER sensor more clearly. We are providing details about the domain structures of the sensor and make it clear the sensor is ER localized and is enriching at ER-PM contacts since it can bind at the same time to the negatively charged lipids unique to the plasma membrane (this is a single-color reporter rather than a FRET sensor). In our analysis, we are normalizing the polarity of the ER-PM contacts relative to a plasma membrane marker (CAAX) or an ER marker (SEC61b).

(ii & iv) Thank you for bringing up the important point about the MTOC orientation. This is important in many migrating cells with likely different roles in amoeboid cells and mesenchymal cells. Mesenchymal cells often migrate with the MTOC and Golgi apparatus in front of the nucleus ('MTOC-front')¹, while amoeboid cells often migrate with the inverted configuration ('MTOC-back') and only reorient once they stably dock onto other cells, as exemplified by immunological synapses²⁻⁴. To investigate the MTOC correlation with the ER-PM contact gradient, we utilized EMTB-mCherry as a marker to label the MTOC and observed its position relative to the nucleus in migrating cells along linear stripes. Surprisingly, we found both types, 'MTOC-front' and 'MTOC-back' in the migrating cells (Fig. R 9a, c, Extended Data Fig. 7 g, i). Interestingly, we found the same stable back-to-front gradient in ER-PM contacts in both cases (Fig. R 9a, b, Extended Data Fig. 7 g, h). Since there is no clear preference of MTOC orientation in the migrating RPE1 cells, MTOC orientation does not seem to be involved in the front-back polarity regulation at least in this system, consistent with the finding by other groups⁵. Nevertheless, our new data in Fig. 5 k, l and Extended Data Fig. 7 e, f now shows in much detail that microtubules are critical in stabilizing ER tubules and sheets.

(iii) We thank the reviewer for appreciating our efforts in testing different cell lines and we apologize if our claim seemed to be an overstatement. We now clarified that our finding is limited to the cell lines we tested. We are cognizant that additional work will be needed to study ER-PM contacts in many more types of migrating cells.

Fig. R9. MTOC position and ER-PM contact gradients in migrating RPE-1 cells. Shown now as Extended Data Fig. 7 g-i. a, (left) Pairs of images captured 60 minutes apart of representative RPE-1 cells stably expressing MAPPER-mVenus (cyan), along with mTurquoise-CaaX (yellow) and EMTB-mCherry (magenta). Right, kymographs of the average back-to-front intensity ratio over time (time interval is 2 min). Positions of MTOC are indicated by the white stars. **b,** Averaged gradients of MAPPER/CaaX ratio from the cell in (a). Mean (dark green) \pm s.e.m. (shaded green). **c,** Percentage of cells with front versus back orientated MTOC position. N= 129 cells from 3 independent experiments.

As the ER is an important site for phosphatidylinositol lipid synthesis, this work might lead to further discoveries in the future about the roles of PIP2 and PIP3 in regulating cell polarization, which seems important but is still rather murky. Perhaps they could add a few more thoughts on this to their sentences on lines 317-319.

Response: We very much agree with the reviewer. This is an important future direction we are interested in investigating. [REDACTED]

Tim Springer

Referee #4:

This study reports a novel mechanism controlling the cell protrusion at the front and suppression of signalling at the back of cell migrating in single or collective migration. Using a live-cell ER-PM reporter, it is shown that it is regulated by PM-ER contact gradients, small and unstable at the front and large, stable ones at the back. It is proposed that ER-PM contact gradients between front and back of the cells shape

the ER morphology (sheets/tubules) through the activity/expression of the two molecules RTN4 and CLIMP63.

How does the ER-PM gradient connect with the actin cytoskeleton /associated molecules to lead to cell protrusion at the front? The authors mention connection with microtubules in the discussion. Could this hypothesis be tested?

Response: The reviewer brings up an interesting point about the connection from ER-PM gradients to microtubules (MT). We do think the ER-microtubule connection is important for the tubular ER formation in the front of migrating cells. In mammalian cells, new ER tubules can be pulled out of the existing ER membrane by associating with motor proteins and then extend along MTs (i.e., the sliding mechanism mediated by the MT-driven motor protein KIF5 and dynein) or, alternatively, by attaching to the tips of polymerizing MTs (i.e., tip attachment complex [TAC] mechanism mediated by the MT end-binding protein 1 [EB1])⁷⁻⁹. Therefore, we hypothesized that KIF5, dynein, and/or EB1 could be needed for generating the ER-PM contact gradients.

To get initial insights which factor is contributing to the process, we transfected cells with siRNAs targeting KIF5B, the subunit p150 of the dynein complex (DCTN1) and EB1 to disrupt the tubular ER-MT connection and then analyzed the distribution of the ER-PM contacts in migrating RPE1 cells. Surprisingly, knockdown of KIF5B greatly reduced the ER-PM gradients (Fig. R 10a, Fig. 5 k), with a very mild enrichment of ER-PM contacts in the back (the back to front ratio is around 1.6 versus 4.6 in the control siRNA group) (Fig. R 10b, Extended Data Fig. 7 e). In contrast, knockdown of DCTN1 did not have a significant effect and EB1 had a smaller effect on the ER-PM gradients, although the absolute enrichment of ER-PM contacts in the back was reduced in EB1 knockdown cells (the back to front ratio was still 2.8 versus 4.6 in the control siRNA group) (Fig. R 10a, b, Extended Data Fig. 7 e). Thus, the KIF5B-mediated tubular ER-MT connection is likely the main driver for the ER-PM gradients in polarized cells with a smaller contribution from EB1.

In addition, CLIMP63, P180 and KTN1, three perinuclear ER-localized sheet ER proteins, have been shown to interact with different microtubule populations to regulate the ER positioning¹⁰, suggesting roles also in the back. We therefore also checked their effects on the ER-PM contact gradients by knocking down their expression. Both CLIMP63 and KTN1 knockdown caused a flattened ER-PM distribution while knockdown of P180 has no significant influence on the ER-PM contact gradients (Fig. R10 c, d, Fig. 5 l, Extended Data Fig. 7 f). This result suggests that the ER sheet protein CLIMP63 or KTN1 might also regulate the ER-PM contact gradients through its microtubule binding ability, since KTN1 knockdown has no effect on the sheet ER morphology¹¹. In summary, this suggests that both the tubular and sheet ER connection with the microtubules contribute to the establishment of the polarized ER-PM contact gradient in the migrating cells.

Fig. R10. Comparison of gradients of MAPPER/CaaX ratio in RPE-1 cells transfected with indicated siRNAs. Shown in Fig. 5 k, l and Extended Data Fig. 7e, f. **a**, Quantification of MAPPER/CaaX profile of RPE-1 cells transfected with control, *KIF5B*, *EB1* or *DCTN1* siRNAs. Mean \pm s.e.m. $n = 46$, control siRNA; $n = 64$, *KIF5B* siRNA; $n = 27$, *EB1* siRNA; $n = 66$, *DCTN1* siRNA. One-way ANOVA (repeated measures) and scheffe's post hoc comparison. *** $p = 0.0005$ (control vs. *KIF5B* siRNA), $p = 0.076$ (control vs. *EB1* siRNA) and $p = 0.4198$ (control vs. *DCTN1* siRNA). **b**, Back to front ratio of MAPPER/CaaX ratio in (a). *** $p < 0.0001$ (control vs. *KIF5B* siRNA), * $p = 0.0479$ (control vs. *EB1* siRNA) and $p = 0.5348$ (control vs. *DCTN1* siRNA). **c**, Same as (a) but cells transfected with control, *P180* or *KTN1* siRNAs. Mean \pm s.e.m. $n = 26$, control siRNA; $n = 73$, *P180* siRNA; $n = 53$, *KTN1* siRNA. One-way ANOVA (repeated measures) and scheffe's post hoc comparison. ns, $p = 0.2075$ (control vs. *P180* siRNA) and * $p = 0.0423$ (control vs. *KTN1* siRNA). **d**, Back to front ratio of MAPPER/CaaX ratio in (c). ns, $p = 0.3272$ (control vs. *P180* siRNA), ** $p = 0.0092$ (control vs. *KTN1* siRNA), ** $p = 0.0062$ (control vs. *CLIMP63* siRNA).

The techniques employed appear mostly appropriate and a number of results are exciting. However, there are a number of issues listed below:

Results displayed Figure 1 are clear but there is no statistics.

Response: Thank you for the positive notes. We are now including the statistical analysis of the data in Fig 1 h, i and Extended Data Fig. 2 d, g. Data are shown as the mean \pm SEM.

Fig. R11. Quantification of gradients of MAPPER/CaaX ratio, MAPPER/ER ratio, MAPPER mass and size in RPE-1 cells. Data are shown as mean \pm s.e.m. Shown in Fig 1 h, i and Extended Data Fig. 2 d, g. **a**, Group average MAPPER/CaaX gradient of 25 cells (black). **b**, Same as (a) for the MAPPER/ SEC61 β ratio. **c,d**, Group averaged gradients of the MAPPER mass (c) or MAPPER size (d) (n=25 cells). Mean (dark green) \pm s.e.m. (shaded green)

Figure 2 panels a-e are too descriptive with only one cell shown with a certain phenotype. These panels need to include quantitative and statistics data. Panel C is not convincing. It is not clear these cells are ongoing collective migration and the gradient of ER-PM is not really visible. Why would it show in only one cell? Panel f and g are quantified but no statistics is provided. Are cells migrating in the amoeboid mode in Panel d? If so, this finding should be expanded as this could mean this PM-ER contact gradient plays a role in mesenchymal and amoeboid migration.

Response: We are now also including quantitative and statistical data of ER-PM contact gradients in the BJ cells, HUVEC cells, as well as RPE1 cells migrating in the 2D or 3D mode. The mean \pm s.e.m data are provided in Fig. 1 n, 1 o, Extended Data Fig. 2j and 2l.

Fig. R12. Quantification of gradients of MAPPER/CaaX ratio in the HUVEC (a), BJ (b) and RPE1 cells migrating in the 3D (c) and 2D (d) mode. Data are shown as mean \pm s.e.m. Shown in Fig. 1 n, 1 o, Extended Data Fig. 2j and 2l.

In the original Figure 2c, we analyze the ER-PM contact gradients in collectively migrating cells by adding a wound healing scratch assay which is defining the collective direction of movement. A time-course of the change of the ER-PM contact gradients (0h, 2h, 4h and 6h after wound) is shown in a group of cells. A representative leader cell and follower cell is shown separately in zoomed image panels in Fig. 1m (Fig. R 5). A statistical analysis is shown in Extended Fig. 2 m,n.

Fig. R5. Distribution of ER-PM contacts in collectively migrating RPE1 cells during wound closure at 0h, 2h, 4h and 6h. Shown in Fig. 1m, Extended Fig. 2 m,n and Supplementary Video 2. **a**, Time-course of the collectively migrating RPE-1 cells stably expressing mTurquoise-CaaX and MAPPER-mVenus in the wound closure assay. A representative leader cell and follower cell are separately zoomed in to reveal the distribution of ER-PM contact gradients across the time. PM are indicated by the white dash boundary. **b,c**, Mean MAPPER/CaaX gradients in leader cells (**b**, $n = 46$ cells) and follower cells (**c**, $n = 33$ cells) during collective cell migration. Mean (dark green) \pm s.e.m. (shaded green).

The statistical data for Fig.2f (now Fig. 2d, Fig. R 13) is provided as the fitted regression line \pm 95% CI.

Fig. R13. Scatter plot of mean MAPPER/CaaX gradient steepness and mean cell speed for individual RPE-1 cells migrating on stripes with fitted regression line \pm 95% CI. Shown in Fig. 2d.

Furthermore, In Fig.2d (now Fig. 1 o), we observed ER-PM gradients in RPE-1 cells migrating in 3D within 0.5 mg/ml collagen matrix. For these conditions, cells are migrating in a manner that is reminiscent of amoeboid migration. We have added a movie to exemplify this type of less persistent cell movement (Fig. R14, Supplementary Video 3).

Fig. R14. MAPPER/CaaX gradient in the RPE1 cells ongoing an amoeboid-like migration. Shown in (Supplementary Video 3).

Figure 4. Quantitative results with statistics should be provided for Fig 4 a-c. The kymographs in d and e present the results of how many cells?

Response: We are now clarifying the statistics of the results in Fig. 4a-c. ER-PM contacts in Fig.4a and b are the representative examples of 99,710 MAPPER puncta from 10 cells moving on stripes. The speed of MAPPER puncta is quantified and shown in Fig. 4g. The cell in Fig.4c is the representative example of 53 cells reversing their migration direction at the end of the stripes. The quantitative data for the cell in Fig.4c is shown in Fig. 4d-f. The kymographs in 4d and e is a representative cell data of 53 cells. Given the high heterogeneity of the paused duration during the turn, we could not average the kymographs of all 53 cells, so we aligned the time-course to the turning timepoint and plotted the MAPPER/CaaX or MAPPER/ER ratio 10 min before and after a turn (Fig. R 15, Extended Fig. 5e). The aligned traces are shown as mean \pm s.e.m.

Fig. R15. Change in MAPPER gradient 10 min before and after a cell takes a turn. Data are shown as mean \pm s.e.m. Shown in Extended Data Fig. 5e.

Figure 5. Although the results presented panels b and c are clearly visible, 5d should provide statistics from 3 independent experiments with at least 30 cells per condition and per experiments analysed.

Response: As requested, we are now also adding statistics instead of showing a representative image. We are representing the data as mean \pm sd from 3 independent experiments. The RTN4/SEC61 β gradient is calculated from 1512 cells from 3 batches (n= 37, 603 and 872 cells), the CLIMP63/SEC61 β gradient is calculated from 962 cells from 3 batches (n=72, 541 and 349 cells).

Fig. R16. Comparison of the RTN4/CLIMP63 ratio, as a measure of ER curvature, to the MAPPER/SEC61 β ratio. Data are shown as mean \pm sd. Shown in Fig. 5d.

The group of Jennifer Lippincott-Schwartz has shown that ER organisation in sheet in tubules is an “old fashion view” of ER morphology and in fact the so-called sheets are packed tubules (Nixon-Abell et al, Science 2016). This manuscript should integrate these findings. Provide some references lines 236 and 237 regarding RTN4 and CLIMP63 roles in the formation of ER tubules and sheets. Is there an explanation why the effect of siCLIMP63 is much stronger than the one of siRTN4? Although the data presented panels e-i are quite convincing, they only suggest the change in ER morphology. The data should be backed up with appropriate imaging of ER.

Response: Thanks for the reminding us of the paper. We are now citing the paper and clarified this point (line 244 and 246). We have modified our description in the lines 235 and 237 (now line 248-250) and references were also added.

There could be several reasons for the observed stronger effect of *siCLIMP63* compared to *siRTN4*:

1) microtubule-dependent tubular ER formation mechanism (such as KTN1, KIF5B mediated tubular ER formation) might compensate by enhancing ER tubule formation, since knockdown of *RTN4* together with *KTN1* or *KIF5B* caused a more severe effect on MAPPER gradient, which have no significant difference with *siCLIMP63* (Fig. R17).

Fig. R17. Quantification of back to front ratio of MAPPER gradient after transfection of indicated siRNA. Data are shown as mean \pm sd. N= 49,68,73,45,72,60 cells for the corresponding group from the left to right. *p =0.0147 (control vs. *RTN4* siRNA), ns, p >0.05.

2) Also, CLIMP63 has been shown to be a centrosomal microtubule binding protein that can change ER positioning, as well as redistributions of other organelles ¹⁰, which might also contribute to the formation of the ER-PM contacts gradient.

3) CLIMP63 could also impact the nuclear positioning through asymmetric nucleo-cytoskeleton connections ¹². So, knockdown of CLIMP63 might affect both the ER-PM contact gradient and nuclear positioning, which may also contribute to the front-back polarity.

We would like to note that we are adding here that the ER morphology and Rtn4 and CLIMP63 distribution are polarized in migrating cells but it is well established that these two proteins play a role in controlling ER morphology: For example, in 2010, the Tom A. Rapoport group compared the luminal width of ER sheets in the control versus CLIMP63 siRNA transfected cells and found a significant decrease from 45-50 nm to 25-30 nm ¹¹. This data is also reproduced by another two groups in 2019 ¹³ and 2022 ¹² by electron microscopy. In 2006, the Tom A. Rapoport group reported the abnormal accumulation of ER sheets in *rtn1/rtn2/yop1* triple KO yeast cells ¹⁴, which is also true in mammalian cell lines, a ~67% increase in ER sheets in *RTN4* KO compared with WT MEFs reported by the William C. Sessa group ¹⁵. Based on these findings, one can plausibly expect that CLIMP63 and *RTN4* have similar roles in shaping the ER morphology in migrating RPE1 cells. Our finding with the *RTN4* knockdown is mostly confirming such an increase in ER sheets after *RTN4* knockdown (Fig. R18). Also, it is worth noting that CLIMP63 knockdown could regulate MAPPER gradient both through decreasing the luminal width of ER sheets ¹¹ and disrupting the sheet ER connection with the microtubules ¹⁰.

Fig. R18. Quantification of sheet ER area after transfection of indicated siRNA. Data are shown as mean \pm sd. N= 11,21,20,20 cells for the corresponding group from the left to right. ****p <0.0001 (control vs. RTN4 siRNA).

Additional comments

- Indicate in the result part what is the “expressed PM marker” (line 72).
- Staining for phospho-tyrosine is not specific of RTK and thus does not determine “the spatial organisation of RTK signalling” per se (line 78).
- Extended data Legend of Figure 1: Details of the “CaaX signal” in RPE-1 cells is missing. Clear specification of the cells must be provided (“expressing a mTurquoise-CaaX PM marker” as in Figure 1 legend).
- Fig 4a. There is a typo in the title “stionary.”

Thank you for pointing these out, we corrected them.

References

- 1 Luxton, G. W. & Gundersen, G. G. Orientation and function of the nuclear-centrosomal axis during cell migration. *Curr Opin Cell Biol* **23**, 579-588, doi:10.1016/j.ceb.2011.08.001 (2011).
- 2 Yoo, S. K. *et al.* The role of microtubules in neutrophil polarity and migration in live zebrafish. *J Cell Sci* **125**, 5702-5710, doi:10.1242/jcs.108324 (2012).
- 3 Eddy, R. J., Pierini, L. M. & Maxfield, F. R. Microtubule asymmetry during neutrophil polarization and migration. *Mol Biol Cell* **13**, 4470-4483, doi:10.1091/mbc.e02-04-0241 (2002).

- 4 Krummel, M. F. & Macara, I. Maintenance and modulation of T cell polarity. *Nat Immunol* **7**, 1143-1149, doi:10.1038/ni1404 (2006).
- 5 Isomursu, A. *et al.* Dynamic Micropatterning Reveals Substrate-Dependent Differences in the Geometric Control of Cell Polarization and Migration. *Small Methods* **8**, e2300719, doi:10.1002/smt.202300719 (2024).
- 6 Funamoto, S., Meili, R., Lee, S., Parry, L. & Firtel, R. A. Spatial and temporal regulation of 3-phosphoinositides by PI 3-kinase and PTEN mediates chemotaxis. *Cell* **109**, 611-623, doi:10.1016/s0092-8674(02)00755-9 (2002).
- 7 Grigoriev, I. *et al.* STIM1 is a MT-plus-end-tracking protein involved in remodeling of the ER. *Curr Biol* **18**, 177-182, doi:10.1016/j.cub.2007.12.050 (2008).
- 8 Waterman-Storer, C. M. & Salmon, E. D. Endoplasmic reticulum membrane tubules are distributed by microtubules in living cells using three distinct mechanisms. *Curr Biol* **8**, 798-806, doi:10.1016/s0960-9822(98)70321-5 (1998).
- 9 Wozniak, M. J. *et al.* Role of kinesin-1 and cytoplasmic dynein in endoplasmic reticulum movement in VERO cells. *J Cell Sci* **122**, 1979-1989, doi:10.1242/jcs.041962 (2009).
- 10 Zheng, P. *et al.* ER proteins decipher the tubulin code to regulate organelle distribution. *Nature* **601**, 132-138, doi:10.1038/s41586-021-04204-9 (2022).
- 11 Shibata, Y. *et al.* Mechanisms determining the morphology of the peripheral ER. *Cell* **143**, 774-788, doi:10.1016/j.cell.2010.11.007 (2010).
- 12 Janota, C. S. *et al.* Shielding of actin by the endoplasmic reticulum impacts nuclear positioning. *Nat Commun* **13**, 2763, doi:10.1038/s41467-022-30388-3 (2022).
- 13 Shen, B. *et al.* Calumenin-1 Interacts with Climp63 to Cooperatively Determine the Luminal Width and Distribution of Endoplasmic Reticulum Sheets. *iScience* **22**, 70-80, doi:10.1016/j.isci.2019.10.067 (2019).
- 14 Voeltz, G. K., Prinz, W. A., Shibata, Y., Rist, J. M. & Rapoport, T. A. A class of membrane proteins shaping the tubular endoplasmic reticulum. *Cell* **124**, 573-586, doi:10.1016/j.cell.2005.11.047 (2006).
- 15 Jozsef, L. *et al.* Reticulon 4 is necessary for endoplasmic reticulum tubulation, STIM1-Orai1 coupling, and store-operated calcium entry. *J Biol Chem* **289**, 9380-9395, doi:10.1074/jbc.M114.548602 (2014).

Reviewer Reports on the First Revision:

Referees' comments:

Referee #1 (Remarks to the Author):

The authors have performed an array of experiments that fully address the points raised in the initial comments. The extended new dataset includes cryo-electron tomograms documenting the density and curvature of contacts at the front and back of cells, optogenetics experiments showing that the direction of cell migration can be reversed by the generation of artificial contacts at the front, and immunostaining of an endogenous ER-PM tether evidencing an increased density of contacts at the back. Additional data further hint at the implication of microtubules in the generation of contact asymmetry, an interesting development that would deserve further publication as the present study is already quite exhaustive.

I congratulate the authors for the comprehensive revision and for the quality of the new data. This is an exquisite work that I have no doubt will be highly cited.

We just spotted one minor error in Fig. 2C which has ticks in fractional units and an axis labeled in %, which incorrectly reads as maximal values of 0.5% instead of 50%.

Referee #2 (Remarks to the Author):

The revised manuscript shows in a refined collection of experiments that migrating cells predominantly form ER-PM contacts on the tail end of a cell. The experiments in Figure 2b elegantly show that the formation of these contacts is responsible for the direction of movement. This comes with more curved ER in the front, while the back ER is more sheet-like, as expected from the distribution of ER-PM contacts. The movement, and the formation of contacts were found to highly depend on ER homeostasis, as ER stress abolished the positive connection between the polarized distribution of ER-PM contacts and movement. The study is timely, exciting and provides a novel mechanism controlling cell movement. The authors have addressed all concerns, as far as I can see and have prepared a concise manuscript that will resonate in the cell biology community.

1. I propose to include the supplementary data with oligomycin, since mitochondrial energy production has been implicated in the movement of metastatic cancer cells. The data presented suggest that this role may not depend on their supply of energy for movement but something else.

Referee #3 (Remarks to the Author):

The authors have responded with new experiments suggested by the reviewers that strengthen the

conclusions. The cryo-electron tomography and optogenetic induction of PM-ER close contacts and demonstration of the same back to front polarity in cells migrating in groups are significant additions. Overall, the story is of wide interest, especially in the cell biology and cell migration fields.

Minor points.

The experiments with Brefeldin and tunicamycin are of limited worth, as these agents have a large number of different effects, particularly tunicamycin which results in misfolding of N-glycosylated proteins.

Referee #4 (Remarks to the Author):

The authors have mostly answered to my questions. There are still the following points remaining:

This question was not answered: How does the ER-PM gradient connect with the actin cytoskeleton /associated molecules to lead to cell protrusion at the front?

I have requested statistical analyses for several figure panels. The authors have responded by adding SEM but no statistic test was performed (for example showing a significant difference of normalized MAPPER/CAAX at back versus front in Fig. 1h). I am unclear why this would not be possible or relevant, can this be clarified.

The description of Nixon-Abell, J. et al. is not correct, please cite it properly. It is important to take in consideration that the dogma sheet/tubules has been challenged upon the use of superresolution microscopy.

Author Rebuttals to First Revision:

Referees' comments:

Referee #1 (Remarks to the Author):

The authors have performed an array of experiments that fully address the points raised in the initial comments. The extended new dataset includes cryo-electron tomograms documenting the density and curvature of contacts at the front and back of cells, optogenetics experiments showing that the direction of cell migration can be reversed by the generation of artificial contacts at the front, and immunostaining of an endogenous ER-PM tether evidencing an increased density of contacts at the back. Additional data further hint at the implication of microtubules in the generation of contact asymmetry, an interesting development that would deserve further publication as the present study is already quite exhaustive.

I congratulate the authors for the comprehensive revision and for the quality of the new data. This is an exquisite work that I have no doubt will be highly cited.

Response: Thanks for your positive comment.

We just spotted one minor error in Fig. 2C which has ticks in fractional units and an axis labeled in %, which incorrectly reads as maximal values of 0.5% instead of 50%.

Response: Thanks for your reminder and we have corrected the y-axis labels.

Referee #2 (Remarks to the Author):

The revised manuscript shows in a refined collection of experiments that migrating cells predominantly form ER-PM contacts on the tail end of a cell. The experiments in Figure 2b elegantly show that the formation of these contacts is responsible for the direction of movement. This comes with more curved ER in the front, while the back ER is more sheet-like, as expected from the distribution of ER-PM contacts. The movement, and the formation of contacts were found to highly depend on ER homeostasis, as ER stress abolished the positive connection between the polarized distribution of ER-PM contacts and movement. The study is timely, exciting and provides a novel mechanism controlling cell movement. The authors have addressed all concerns, as far as I can see and have prepared a concise manuscript that will resonate in the cell biology community.

Response: Thanks for your positive comment.

1. I propose to include the supplementary data with oligomycin, since mitochondrial energy production has been implicated in the movement of metastatic cancer cells. The data presented suggest that this role may not depend on their supply of energy for movement but something else.

Response: Thanks for your point and we have added the oligomycin data into Extended Data Figure 8 g-i.

Referee #3 (Remarks to the Author):

The authors have responded with new experiments suggested by the reviewers that strengthen the conclusions. The cryo-electron tomography and optogenetic induction of PM-ER close contacts and demonstration of the same back to front polarity in cells migrating in groups are significant additions. Overall, the story is of wide interest, especially in the cell biology and cell migration fields.

Response: Thanks for your positive comment.

Minor points.

The experiments with Brefeldin and tunicamycin are of limited worth, as these agents have a large number of different effects, particularly tunicamycin which results in misfolding of N-glycosylated proteins.

Response: Thanks for your kind reminder and we also realized the limitation of these inhibitors data, so in the result part, we don't make any concrete conclusion based on these observations. Now some discussions related to that part are included as well.

Referee #4 (Remarks to the Author):

The authors have mostly answered to my questions. There are still the following points remaining:

This question was not answered: How does the ER-PM gradient connect with the actin cytoskeleton /associated molecules to lead to cell protrusion at the front?

Response: We apologize for not answering this question clearly and it is an interesting point to think about the connection between the ER-PM contact with the actin cytoskeleton, which is part of our on-going project. As reported by the Jen Liou group (PMID:28954864) in 2017, most of ER-PM contacts were not superimposed on, but in close proximity to F-actin. And we confirmed this observation in the migrating RPE1 cells, indicating a spatial coordination of ER-PM contacts and cortical actin. And indeed, in 2020, our group discovered that in the back of migrating cells, similar to ER-PM contact gradients, there was an enrichment of membrane-proximal F-actin (MPAct gradient), which is parallel to the plasma membrane and restricts local membrane protrusion (PMID:32527825). Therefore, we think both the ER-PM contact gradient and MPAct gradient are required to direct cell protrusion at the front.

[REDACTED]

I have requested statistical analyses for several figure panels. The authors have responded by adding SEM

but no statistic test was performed (for example showing a significant difference of normalized MAPPER/CAAX at back versus front in Fig. 1h). I am unclear why this would not be possible or relevant, can this be clarified.

Response: We apologize for missing this statistic test and now the front-to-back significant difference tests have been added.

The description of Nixon-Abell, J. et al. is not correct, please cite it properly. It is important to take in consideration that the dogma sheet/tubules has been challenged upon the use of superresolution microscopy.

Response: Thanks for your comment and we have revised the description as “The ER has been shown to have a dynamic and complex morphology that includes tubules with diameters of about 60 nm being more peripheral localized, and flat sheet-like structures that are often more perinuclear.” We would like to clarify that we are not addressing this interesting complexity of these ER morphology, but rather use two different approaches, Cryo-ET to measure ER curvature and light microscopy to measure the ratio of RTN4 (curved ER membrane) and CLIMP63 (flatten ER membrane). Both approaches have demonstrated a polarized ER curvature gradient in the migrating cells, which is closely correlated with the formation of ER-PM contact gradients. We have changed some of descriptions in the main text.